# Iterative Self-Incentivization Empowers Large Language Models as Agentic Searchers

**Zhengliang Shi**[1]    **Lingyong Yan**[2]    **Dawei Yin**[2]
**Suzan Verberne**[3]    **Maarten de Rijke**[4]    **Zhaochun Ren**[3]*
[1]Shandong University, Qingdao, China  [2]Baidu. Inc, Beijing, China
[3]Leiden University, Leiden, The Netherlands
[4]University of Amsterdam, Amsterdam, The Netherlands
{zhengliang.shii, lingyongy}@gmail.com  yindawei@acm.org
{s.verberne, z.ren}@liacs.leidenuniv.nl  M.deRijke@uva.nl

## Abstract

Large language models have been widely integrated into information retrieval to advance traditional techniques. However, effectively enabling LLMs to seek accurate knowledge in complex tasks remains a challenge due to the complexity of multi-hop queries as well as the irrelevant retrieved content. To address these limitations, we propose EXSEARCH, an agentic search framework, where the LLM learns to retrieve useful information as the reasoning unfolds through a self-incentivized process. At each step, the LLM decides what to retrieve (*thinking*), triggers an external retriever (*search*), and extracts fine-grained evidence (*recording*) to support next-step reasoning. To enable LLM with this capability, EXSEARCH adopts a generalized expectation-maximization algorithm. In the E-step, the LLM generates multiple search trajectories and assigns an importance weight to each; the M-step trains the LLM on them with a re-weighted loss function. This creates a self-incentivized loop, where the LLM iteratively learns from its own generated trajectories, progressively improving itself for search. We theoretically analyze this training process and establish convergence guarantees. Extensive experiments on four benchmarks show that EXSEARCH outperforms baselines substantially, e.g., +7.8% improvement on exact match score. Motivated by these promising results, we introduce EXSEARCH-Zoo, an extension that extends our method to broader scenarios, to facilitate future work. Code is available on ⌗ EXSEARCH.

## 1  Introduction

Information Retrieval (IR), one of the most fundamental data mining techniques, aims to understand complex queries and extract relevant information from external sources for users [15, 34]. Recently, large language models (LLMs), which exhibit remarkable language understanding and reasoning abilities, have been widely integrated into IR to enhance traditional techniques [22, 89]. For example, some work uses LLMs for query rewriting [61] or document re-ranking [67] to augment search engines, while other work uses LLMs to summarize information and generate accurate responses [78].

Despite the progress made by LLMs in IR applications, enabling LLMs to effectively seek accurate knowledge in complex downstream tasks remains a challenge [4, 16, 72]. Specifically, many real-world tasks, such as multi-hop question answering, require iterative and dynamic retrieval, where directly issuing a complex query composed of multiple sub-queries often results in low retrieval coverage [30, 33]. As retrieval is typically imperfect [82, 87], even for a simple one-hop query, the retrieved results contain irrelevant content, leading to a misleading context [4, 88]. Thus, it is crucial to teach LLMs how to interact with retrievers and reflect on retrieved content as reasoning unfolds.

---

* Corresponding author.

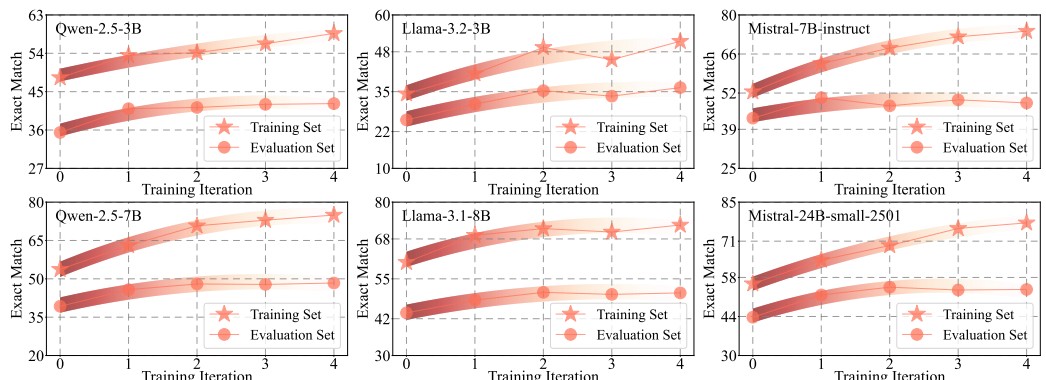

Figure 1: Performance on HotpotQA dataset when applying our EXSEARCH to different LLMs.

To address these limitations, previous work typically cascades information-seeking pipelines (e.g., query decomposition [43] or knowledge extraction [52]), and trains LLMs for specific stages with synthetic data independently [40, 88]. However, aligning different retrieval stages with end-to-end supervision remains an active research area.

In this paper, we propose EXSEARCH, an **ex**ploratory **search** framework that empowers an LLM as an agent with fine-grained reasoning and search capabilities through a self-incentivized process. Specifically, EXSEARCH formalizes three core actions: (i) *thinking*: generating a query based on the evolving search trajectory; (ii) *search*: triggering a retriever; and (iii) *recording*: extracting fine-grained evidence to support subsequent reasoning. Given an input task, the LLM interleaves these actions to progressively explore a search trajectory, which is sequentially composed of sub-queries, retrieved documents, and supporting evidence. The final answer is generated based on the entire trajectory, effectively grounding the generated response in external knowledge.

To incentivize the LLM with this agentic search, EXSEARCH adopts the generalized expectation-maximization (GEM) algorithm, which alternates between two main steps: *trajectory exploration* (E-step) and *re-weighted trajectory learning* (M-step). The key innovation is treating search trajectories as latent variables, training the LLM to reason and search end-to-end. In the E-step, we approximate the current distribution over search trajectories through importance sampling [69]. For each input task, the LLM generates candidate trajectories, each automatically assigned an importance weight based on how well it supports the correct answer. In the M-step, we reweight these trajectories to construct an evidence lower bound (ELBO), which is then maximized to update the LLM parameters. This step enables the LLM to learn from its own generated data, encouraging it to generate more supportive trajectories and accurate answers. By interleaving the EM steps, we form a self-incentivized loop that progressively optimizes the model to search relevant knowledge and reason over it.

Our method tightly integrates LLM with IR and diverges from prior work by enabling a unified LLM for dynamic document retrieval, evidence extraction, and answer aggregation through a self-improving process. Extensive experiments on a wide range of knowledge-intensive benchmarks demonstrate the improvement of EXSEARCH over strong baselines. To further understand its advantages, we provide a theoretical analysis, which illustrates that the proposed self-incentivized framework ensures stable convergence, as briefly shown in Figure 1. These promising results motivate us to extend EXSEARCH to more diverse scenarios. We therefore introduce EXSEARCH-Zoo, a comprehensive resource that extends EXSEARCH by two dimensions: (i) diverse backbone LLMs across different model families (LLaMA, Qwen) and scales (7B, 24B parameters); and (ii) extended actions, i.e., document re-ranking, to enrich the action space in vanilla EXSEARCH.

Our contributions are summarized as follows:

1. We propose EXSEARCH, an exploratory search framework that empowers an LLM for agentic retrieval, evidence extraction, and answer aggregation through a self-incentivization process.

2. We provide theoretical analysis on our training process and establish convergence guarantees. Extensive experiments on four benchmarks also demonstrates the superiority of our method.

3. We introduce EXSEARCH-Zoo, an extended resource that generalizes EXSEARCH to more scenarios, facilitating future research.

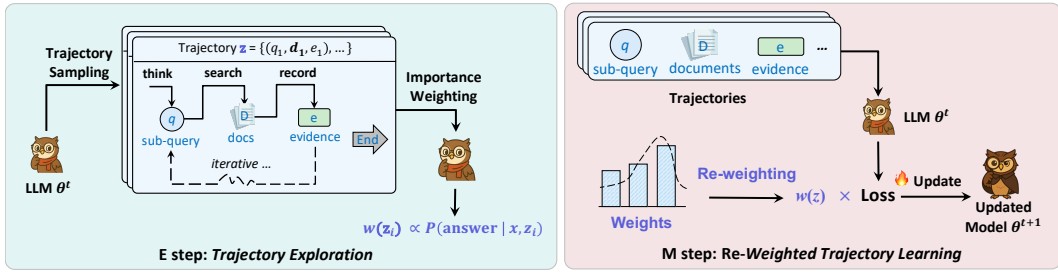

Figure 2: Overall framework of the Expectation-Maximization process in EXSEARCH. The E-step samples search trajectories and assigns each a weight based on its likelihood leading to a correct answer. The special token End marks the completion of reasoning. The M-step trains the LLM, encouraging the LLM to generate more supportive search trajectories and accurate answers.

## 2 Method: EXSEARCH

In this section, we first introduce how the proposed method, EXSEARCH, models the iterative and dynamic search strategy as an agentic *information-seeking* [6, 92] process. Next, we explain how the LLM-based search agent in EXSEARCH can be incentivized to learn this search pattern through a self-improving process. The theoretical analysis of this learning process is presented in § 3.

### 2.1 Search process in EXSEARCH

Our method draws inspiration from *Exploratory Search* [44], a well-established paradigm that conceptualizes searching as a dynamic, iterative process, involving the continuous refinement of queries based on retrieved intermediate results. Accordingly, EXSEARCH simulates this process through an agentic flow, interleaving several actions: *thinking*, *search*, and *recording*, which progressively explore a search trajectory. Specifically, in *thinking*, EXSEARCH determines the next retrieval action by formulating a query $x_i$ based on the current trajectory. In *search*, an external retrieval module $\mathcal{R}$ is triggered to obtain the top-$K$ relevant documents $\boldsymbol{d}_i = \mathcal{R}(x_i) = d_{i,j} \mid j \in [K]$ for the query $x_i$. Finally, in *recording*, EXSEARCH reflects on the retrieved documents and extracts fine-grained evidence $e_i$ to support the subsequent retrieval stage. Formally, we model the LLM-explored search trajectory $\boldsymbol{z}$ as a sequence $\boldsymbol{z} = \{(x_i, \boldsymbol{d}_i, e_i) \mid i \in [|\boldsymbol{z}|]\}$, with a joint likelihood:

$$p(\boldsymbol{z} \mid x; \theta) = \prod_{i=1}^{|\boldsymbol{z}|} p\left((x_i, \boldsymbol{d}_i, e_i)|x, \boldsymbol{z}_{<i}; \theta\right) = \prod_{i=1}^{|\boldsymbol{z}|} p\left(x_i \mid x, \boldsymbol{z}_{<i}; \theta\right) \cdot p\left(e_i \mid x_i, \mathcal{R}(x_i); \theta\right), \text{ (1)}$$

where the $\theta$ denotes the LLM parameters. After such an interleaved search and reasoning process, the LLM generates the final answer $y$ by aggregating the information from $\boldsymbol{z}$, denoted as: $y \sim p(y \mid x, \boldsymbol{z}; \theta)$. The system prompt used in EXSEARCH is provided in Appendix F.1.

### 2.2 Iterative self-incentivization for EXSEARCH

Figure 2 presents the overall framework. EXSEARCH empowers the LLM with agentic search capabilities by alternating between the E-step and M-step, based on the generalized expectation-maximization algorithm [45]. Below, we begin by deriving the training objective of EXSEARCH. We then detail the iterative training process, demonstrating how the LLM progressively improves through trajectories generated by itself.

**Preliminary: Deriving the learning objective.** Given the user's input task $x$, we define the training objective in EXSEARCH as optimizing the LLM to generate the correct answer $y$ after gathering useful information, which can be modeled as the joint probability of the search and generation process:

$$\log p(y \mid x; \theta) = \log \sum_{\boldsymbol{z}} p(y, \boldsymbol{z} \mid x; \theta). \tag{2}$$

Here $\boldsymbol{z} = \{(x_i, \boldsymbol{d}_i, e_i) \mid i \in [|\boldsymbol{z}|]\}$ which denotes a trajectory interleaving the *thinking*, *search* and *recording* introduced in § 2.1. However, marginalizing over all possible trajectories $\boldsymbol{z}$ is generally intractable. To address this, we treat $\boldsymbol{z}$ as a latent variable and derive a variational lower bound. Specifically, we introduce a proposal distribution $q(\boldsymbol{z} \mid x)$ to estimate the sampling space of $\boldsymbol{z}$ and

apply *Jensen's inequality* to the marginal log-likelihood in Eq. (2):

$$\log \sum_{\boldsymbol{z}} q(\boldsymbol{z}|x) \frac{p(y, \boldsymbol{z}|x; \theta)}{q(\boldsymbol{z}|x)} \geq \sum_{\boldsymbol{z}} q(\boldsymbol{z}|x) \log \frac{p(y, \boldsymbol{z}|x; \theta)}{q(\boldsymbol{z}|x)} = \mathbb{E}_{\boldsymbol{z} \sim q(\boldsymbol{z}|x)} \left[ \log \frac{p(y, \boldsymbol{z}|x; \theta)}{q(\boldsymbol{z}|x)} \right]. \quad (3)$$

We refer to this expectation $\mathbb{E}_{\boldsymbol{z} \sim q(\boldsymbol{z}|x)}[\log \frac{p(y, \boldsymbol{z}|x; \theta)}{q(\boldsymbol{z}|x)}]$ as the variational evidence lower bound (ELBO) of $\log p(y|x; \theta)$. The bound $(\geq)$ becomes tight if and only if the proposal distribution matches the true posterior: $q(\boldsymbol{z}|x) \approx p(\boldsymbol{z} \mid y, x; \theta)$. Therefore, we can naturally improve the $\log p(y|x; \theta)$ from the perspective of Expectation-Maximization. In each iteration, the E-step estimates the distribution $p(\boldsymbol{z} \mid y, x; \theta)$ over trajectory $\boldsymbol{z}$ while the subsequent M-step updates the LLM parameters $\theta$ by maximizing the ELBO under the sampled trajectories.

**E-step: Trajectory exploration.** In the $t-$th training, to achieve the boundedness of Eq. (3), we assume $q(\boldsymbol{z}|x) \approx p(\boldsymbol{z} \mid x, y; \theta^t)$ and rewrite the ELBO as follows. The $\mathcal{H}(\cdot)$ denotes the entropy.

$$\text{ELBO} = \mathbb{E}_{\boldsymbol{z} \sim p(\boldsymbol{z}|x, y; \theta^t)} \left[ \log p(y, \boldsymbol{z} \mid x; \theta) \right] + \underbrace{\mathcal{H}(p(\boldsymbol{z} \mid x, y; \theta^t))}_{\text{entropy of } p(\boldsymbol{z} \mid x, y; \theta^t)}. \quad (4)$$

The entropy $\mathcal{H}(p(\boldsymbol{z}|x; \theta^t))$ is non-related to $\theta$ and can be denoted as a constant $c$. As pointed by prior work [58, 69, 93], sampling from the exact posterior (e.g., $p(\boldsymbol{z}|x, y; \theta^t)$) is typically intractable. Therefore, we apply an *importance sampling* strategy [14, 69]. The main conception is instead sampling from a simple distribution (i.e., $p(\boldsymbol{z}|x; \theta^t)$ in this work), and assigning each sample with an importance weight. This corresponds to producing the trajectory $\boldsymbol{z}$ as formulated in Eq. (1) by the current LLM with parameters $\theta^t$. With this strategy, we rewrite Eq. (4) as:

$$\text{ELBO} = \mathbb{E}_{\boldsymbol{z} \sim p(\boldsymbol{z}|x; \theta^t)} \left[ w(\boldsymbol{z}) \log p(y, \boldsymbol{z} \mid x; \theta) \right] + c, \quad (5)$$

where $w(\boldsymbol{z}) := p(\boldsymbol{z} \mid x, y; \theta^t)/p(\boldsymbol{z} \mid x; \theta^t)$ denotes the importance weight. Using Bayes' theorem, we can decompose the posterior as: $p(\boldsymbol{z} \mid x, y; \theta^t) \propto p(\boldsymbol{z} \mid x; \theta^t) \times p(y \mid x, \boldsymbol{z}; \theta^t)$, which allows the importance weight to be further simplified as $w(\boldsymbol{z}) \propto p(y \mid \boldsymbol{z}, x; \theta^t)$. This weight directly reflects how well the search trajectory $\boldsymbol{z}$ supports the LLM in generating the correct answer $y$.

**M-step: Re-weighted trajectory learning.** In the M-step, we update the model parameters $\theta$ by maximizing the ELBO in Eq. (5), formulated as: $\theta = \arg\max_\theta \text{ELBO}$. By applying the product rule to decompose the joint likelihood $p(y, \boldsymbol{z}|x; \theta)$, we can split the ELBO into two terms as follows, which corresponds to two objectives: (i) learning to search via $\mathcal{L}_\mathcal{R}$; (ii) learning to answer via $\mathcal{L}_\mathcal{A}$.

$$\theta = \arg\max_\theta \mathbb{E}_{\boldsymbol{z} \sim p(\boldsymbol{z}|x; \theta^t)} \left[ w(\boldsymbol{z}) \log p(y, \boldsymbol{z} \mid x; \theta) \right]$$
$$= \arg\max_\theta \mathbb{E}_{\boldsymbol{z} \sim p(\boldsymbol{z}|x; \theta^t)} \left[ \underbrace{w(\boldsymbol{z}) \log p(\boldsymbol{z} \mid x; \theta)}_{\text{learning to reason}} + \underbrace{w(\boldsymbol{z}) \log p(y \mid x, \boldsymbol{z}; \theta)}_{\text{learning to answer}} \right]. \quad (6)$$

Therefore, we can understand and interpret the training process in the M-step from both retrieval and answer aggregation aspects. In more details, the term $\mathcal{L}_\mathcal{R} := w(\boldsymbol{z}) \log p(\boldsymbol{z} \mid x; \theta)$ encourages the model to generate high-quality search trajectory $\boldsymbol{z} = \{(x_i, \boldsymbol{d}_i, e_i) \mid i \in [|\boldsymbol{z}|]\}$ to gather relevant documents. According to Eq. 1, $\mathcal{L}_\mathcal{R}$ can be calculated as:

$$\mathcal{L}_\mathcal{R} = \sum_{i=0}^{|\boldsymbol{z}|} \log p(x_i \mid x, \boldsymbol{z}_{<i}; \theta) + \log p(e_i \mid \boldsymbol{d}_i, x_i; \theta), \quad (7)$$

where $\boldsymbol{z}_{<i} = \{(x_{<i}, \boldsymbol{d}_{<i}, e_{<i})\}$ denotes the trajectory up to $i$th step. Similarly, the $\mathcal{L}_\mathcal{A} := \log p(y \mid x, \boldsymbol{z}; \theta)$ trains the model to aggregate all useful information from $\boldsymbol{z}$ to produce the final answer. The overall optimization in this M-step can be achieved via stochastic gradient descent, which is highly compatible to existing computation library like Pytorch. The gradient $\nabla_\theta$ with respect to $\theta$ can be computed as:

$$\nabla_\theta \text{ELBO}(\theta) = -\mathbb{E}_{\boldsymbol{z} \sim p(\boldsymbol{z}|x; \theta^i)} \left[ w(\boldsymbol{z}) \nabla_\theta (\mathcal{L}_\mathcal{R} + \mathcal{L}_\mathcal{A}) \right]. \quad (8)$$

**Pseudo-algorithm.** To clarify the overall procedure in EXSEARCH, we present a pseudo-algorithm in Algorithm 1. In the E-step, the LLM generates search trajectories on its own and evaluates each one with an importance weight $w(\boldsymbol{z})$, reflecting how well the trajectory supports generating the correct answer. In the M-step, the LLM is trained on these trajectories using a weighted loss, learning to generate supportive search paths and accurate answers.

**Algorithm 1:** Training process in EXSEARCH, which alternates between the E-step and M-step.

---

**Input:** Initial LLM $\theta^0$; Training data $\mathcal{D} = \{(x^i, y^i)\}_{i=1}^N$; Training iteration $N$; Maximal step $T$.

**for** $t = 0$ *to* $N$ **do**

    // E-step: Trajectory Exploration

    **for** *example* $(x, y)$ **in** *training set* **do**

        **while** *no end and* $i < T$ **do**

            Generate sub-query $x_i \sim p(x_i \mid x, \boldsymbol{z}; \theta^t)$    ▷ thinking: generate sub-query

            Retrieve document candidates $\boldsymbol{d}_i = \mathcal{R}(x_i)$    ▷ search: retrieve documents

            Extract evidence $e_i \sim p(e_i \mid x_i, \boldsymbol{d}_i; \theta^t)$    ▷ recording: extract key evidence

            $\boldsymbol{z} \leftarrow \boldsymbol{z} \cup (x_i, \boldsymbol{d}_i, e_i)$    ▷ append actions into trajectory

        Compute importance weight: $w(\boldsymbol{z}) = p(y \mid x, \boldsymbol{z}; \theta^t)$

    // M-step: Re-Weighted Trajectory Learning

    $\mathcal{L}_{\mathcal{R}}(x, y; \theta) := \mathbb{E}_{\boldsymbol{z} \sim p(\boldsymbol{z}|x; \theta^t)} [w(\boldsymbol{z}) \log p(\boldsymbol{z} \mid x; \theta)]$    ▷ define the loss for reasoning

    $\mathcal{L}_{\mathcal{A}}(x, y; \theta) := \mathbb{E}_{\boldsymbol{z} \sim p(\boldsymbol{z}|x; \theta^t)} [w(\boldsymbol{z}) \log p(y \mid x, \boldsymbol{z}; \theta)]$    ▷ define the loss for answer

    $\theta^{t+1} = \arg\max_\theta \mathbb{E}_{(x,y) \sim \mathcal{D}} [\mathcal{L}_{\mathcal{R}}(x, y; \theta) + \mathcal{L}_{\mathcal{A}}(x, y; \theta)]$    ▷ optimize through gradient

    **if** *no improvement on validation set* **then**

        Stop training    ▷ Early Stop

**Output:** $\theta$

---

## 3 Theoretical Analysis

During the training of EXSEARCH, we alternate between the E-step and M-step to progressively improve the LLM. Below, we prove the training convergence by showing the non-decreasing optimization after each iteration and analyzing the upper-bounded property of the learning objective.

**Lemma 3.1** (Non-decreasing optimization). *After training in the $t$th ($t \in \mathbb{Z}^+$) iteration, the overall learning objective* $\log p(y \mid x, \theta^{t+1})$ *satisfies* $\log p(y \mid x, \theta^{t+1}) \geq \log p(y \mid x, \theta^t)$.

*Proof.* At $t$th iteration, we start with an E-step to sample trajectories $\boldsymbol{z}$ from the current LLM $\theta^t$ via an importance sampling strategy. As described in Eq. (5), the ELBO can be written as:

$$\text{ELBO}(\theta, \theta^t) := \mathbb{E}_{\boldsymbol{z} \sim p(\boldsymbol{z}|x; \theta^t)} [w(\boldsymbol{z}) \log p(y, \boldsymbol{z} \mid x; \theta)] + c.$$

In the subsequent M-step, we update $\theta$ by maximizing the ELBO: $\theta^{t+1} = \arg\max_\theta \text{ELBO}(\theta, \theta^t)$. By construction, this ensures $\text{ELBO}(\theta^{t+1}, \theta^t) \geq \text{ELBO}(\theta^t, \theta^t)$. Furthermore, using the definition of ELBO from Eq. (3), we observe:

$$\log p(y \mid x; \theta^{t+1}) \geq \text{ELBO}(\theta^{t+1}, \theta^t) \geq \text{ELBO}(\theta^t, \theta^t) = \log p(y \mid x; \theta^t). \tag{9}$$

Thus, we establish that $\log p(y|x; \theta^{t+1}) \geq \log p(y|x; \theta^t)$, demonstrating the non-decreasing nature of the optimization process. Next, we analyze the boundedness of the learning objective. Since $p(y|x; \theta) \in [0, 1]$, it follows $\log p(y|x; \theta) \in (-\infty, 0]$. Therefore, the sequence $\{\log p(y|x; \theta^t)\}_{t=1}^\infty$ is non-decreasing and upper-bounded. We then apply the following convergence theorem from [5]:

**Theorem 3.2** (Monotone convergence theorem). *If a sequence $\{a_n\}$ is monotonic (either non-decreasing or non-increasing) and bounded, then it converges to a finite limit.*

Applying Theorem 3.2, the non-decreasing and upper-bounded nature of $\{\log p(y|x; \theta^t)\}_{t=1}^\infty$ ensures that the sequence converges to a finite value, proving the training convergence of EXSEARCH. We also provide more analysis in Appendix C.3 to rethink and understand the training process. $\qquad\square$

## 4 Experimental Setup

**Datasets and evaluation metrics.** Following prior work [37, 57, 80, 88, 90], we conduct experiments on a range of well-established benchmarks: Natural Questions (NQ) [36], HotpotQA [84], MuSiQue [70], and 2WikiMultihopQA (2WikiQA) [20]. Table 5 in Appendix D summarizes their key statistics. We use three metrics from KILT [49]: *F1*, *Exact Match* (EM), and *Accuracy* (Acc).

Table 1: Comparison between our proposed EXSEARCH and baselines, with the **best** and **second best** results in bold. Blue lines represent the results of EXSEARCH using the same backbone LLMs as the most commonly-used baselines. **Green lines** represent analysis results for applying EXSEARCH to various LLMs. − indicates no open-source code can be used to run the experiments. * indicates cases where the model produces long-form answers, which struggle to align with the short-span ground truth format. In these cases, we suggest using Acc. as a more reliable metric.

| Tasks | NQ | | | HotpotQA | | | MuSiQue | | | 2WikiQA | | | Avg. | | |
|---|---|---|---|---|---|---|---|---|---|---|---|---|---|---|---|
| **Metrics** | F1 | EM | Acc. | F1 | EM | Acc. | F1 | EM | Acc. | F1 | EM | Acc. | F1 | EM | Acc. |
| *Direct reasoning without retrieval-augmented generation* | | | | | | | | | | | | | | | |
| Deepseek-R1-671B [17] | 49.45 | 35.71 | 43.83 | 46.98 | 35.83 | 37.80 | 17.34 | 10.22 | 12.69 | 52.18 | 43.83 | 50.66 | 41.49 | 31.40 | 36.24 |
| GPT-4o [48] | 48.76 | 35.75 | 43.03 | 54.13 | 36.52 | 51.59 | 29.07 | 18.92 | 22.97 | 51.31 | 40.45 | 53.07 | 45.82 | 32.90 | 42.66 |
| GPT-3.5-turbo [47] | 42.11 | 38.60 | 40.60 | 34.90 | 24.57 | 31.86 | 22.73 | 14.14 | 16.29 | 33.90 | 30.40 | 32.45 | 33.41 | 26.93 | 30.30 |
| Qwen2.5-72B | 45.68 | 30.12 | 45.67 | 38.80 | 29.20 | 32.00 | 20.40 | 11.40 | 12.54 | 42.70 | 34.40 | 33.61 | 36.90 | 26.28 | 30.96 |
| Llama-3.3-70B [13] | 48.70 | 36.00 | 36.00 | 49.10 | 37.80 | 39.20 | 23.60 | 14.80 | 16.00 | 54.20 | 46.00 | 50.60 | 43.90 | 33.65 | 35.45 |
| Mistral-8x7B [25] | 40.87 | 40.10 | 39.60 | 25.19 | 25.80 | 25.80 | 11.60 | 6.80 | 14.80 | 30.21 | 27.05 | 29.67 | 26.97 | 22.59 | 27.47 |
| QwQ-32B [68] | 33.09 | 23.00 | 32.20 | 33.34 | 25.40 | 26.60 | 18.85 | 9.00 | 9.00 | 40.90 | 34.40 | 36.40 | 31.55 | 22.95 | 26.05 |
| Qwen-2.5-32B [83] | 33.10 | 23.00 | 34.20 | 34.74 | 25.40 | 27.60 | 18.90 | 8.50 | 10.03 | 36.34 | 29.80 | 34.40 | 30.77 | 21.68 | 26.58 |
| *Advanced retrieval-augmented generation* | | | | | | | | | | | | | | | |
| RankRAG (llama-3.1-70B) [88] | – | **54.20** | – | 55.40 | 42.70 | – | – | – | – | 43.90 | 38.20 | – | – | – | – |
| ChatQA (llama-2-70B) [42] | 34.54 | 23.64 | 37.41 | 44.60 | 33.40 | 33.40 | 17.05 | 16.64 | 19.24 | 31.90 | 26.80 | 32.56 | 32.02 | 25.12 | 30.65 |
| Recomp (Flan-UL2-20B) [78] | 42.67 | 37.47 | 40.32 | 42.72 | 38.72 | 41.55 | 24.96 | 17.34 | 21.46 | 38.26 | 32.17 | 36.12 | 37.15 | 31.43 | 34.86 |
| Retrobust (llama-2-13B) [87] | 43.82 | 37.03 | 39.56 | 40.54 | 35.59 | 38.79 | 18.16 | 18.11 | 19.61 | 39.11 | 38.65 | 39.77 | 35.41 | 32.34 | 34.43 |
| InstructRAG (llama-3.1-8B) [74] | 39.21 | 37.82 | 37.58 | 37.31 | 36.77 | 35.31 | 25.88 | 14.94 | 20.45 | 40.01 | 44.57 | 38.91 | 35.60 | 33.52 | 33.06 |
| RAG-DDR (llama-3.1-8B) [40] | 40.74 | 28.76 | 30.51 | 31.71 | 40.04 | 42.41 | 13.54 | 10.57 | 14.21 | 38.40 | 35.44 | 37.41 | 31.10 | 28.70 | 31.14 |
| RankRAG (llama-3.1-8B) [88] | – | **50.60** | – | 46.70 | 35.30 | – | – | – | – | 36.90 | 31.40 | – | – | – | – |
| ChatQA (llama-2-7B) [42] | 34.54 | 23.64 | 37.41 | 44.60 | 33.40 | 33.40 | 17.05 | 16.64 | 19.24 | 31.90 | 26.80 | 32.56 | 32.02 | 25.12 | 30.65 |
| *Iterative retrieval-augmented generation* | | | | | | | | | | | | | | | |
| GenGround (GPT-3.5) [64] | 50.31 | 40.24 | 43.60 | 52.26 | 45.31 | 47.27 | 27.36 | 18.34 | 20.24 | 50.21 | 42.31 | 45.61 | 45.04 | 36.55 | 39.18 |
| IRCoT (GPT-3.5) [71] | 45.42 | 42.41 | 43.21 | 58.40 | 45.50 | 46.32 | **30.50** | 19.01 | 22.87 | 45.10 | 35.40 | 36.54 | 44.86 | 35.58 | 37.24 |
| DSPy (GPT-3.5) [32] | 42.25 | 29.10 | 42.00 | 47.10 | 34.67 | 42.73 | 19.88 | 10.80 | 13.40 | 44.52 | 39.64 | 44.43 | 38.44 | 28.55 | 35.64 |
| SearChain (GPT-3.5) [80] | 8.25* | 0.00* | 45.43 | 6.18* | 0.00* | 47.64 | 2.51* | 0.00* | 9.22 | 6.05* | 0.00* | 43.69 | 5.75* | 0.00* | 36.49 |
| Iter-RetGen (GPT-3.5) [61] | 28.30 | – | 41.04 | 44.10 | – | 21.04 | 17.69 | – | 20.19 | 36.00 | – | 42.17 | 31.52 | – | 31.11 |
| Verify-and-Edit (GPT-3.5) [94] | 39.73* | 26.68* | 40.34 | 12.44* | 0.00* | 27.43 | 5.87* | 0.00* | 10.01 | 13.39* | 0.00* | 32.68 | 17.86* | 6.67* | 27.62 |
| Gen-Ret-Gen (GPT-3.5) [1] | 46.66 | 38.06 | 48.88 | 49.59 | 37.69 | 45.03 | 25.94 | 13.24 | 17.82 | 40.26 | 29.43 | 39.23 | 40.61 | 29.60 | 37.73 |
| Search-o1 (QwQ-32B) [39] | 47.52 | 32.41 | 40.34 | 53.31 | 43.51 | 45.31 | 25.41 | 16.64 | 19.42 | 50.31 | 42.61 | 45.41 | 44.14 | 33.79 | 37.62 |
| Search-R1 (Qwen-2.5-7B) [28] | 54.26 | 42.21 | **51.35** | 58.04 | 46.51 | 50.03 | 21.21 | **23.37** | **25.62** | 49.64 | **50.43** | 48.74 | | 39.89 | 44.02 |
| SELF-RAG (llama-2-7B) [4] | 49.70 | 41.60 | 42.50 | 21.50 | 9.40 | 29.20 | 21.50 | 9.43 | 7.10 | 27.33 | 23.52 | 20.80 | 30.01 | 20.99 | 24.90 |
| Ours-Qwen-2.5-7B | **56.37** | 47.07 | **51.75** | **62.59** | **50.35** | **54.32** | 29.68 | **22.03** | 24.34 | **57.14** | **52.62** | **54.37** | **51.45** | **43.02** | **46.20** |
| Ours-Llama-3.1-8B | **55.21** | 43.71 | 50.76 | **60.72** | **47.59** | **53.59** | **30.83** | 20.98 | **24.65** | **54.62** | 47.48 | **54.21** | **50.35** | **39.94** | **45.80** |
| *Extended analysis on various backbone LLMs* | | | | | | | | | | | | | | | |
| Ours-Qwen-2.5-3B | 46.23 | 36.76 | 39.12 | 54.32 | 42.22 | 46.08 | 19.44 | 13.76 | 13.94 | 43.39 | 37.24 | 44.78 | 40.85 | 32.50 | 35.98 |
| Ours-Llama-3.3-3B | 41.42 | 33.49 | 35.17 | 44.12 | 33.53 | 36.14 | 17.64 | 11.23 | 11.82 | 41.73 | 36.28 | 42.22 | 36.23 | 28.63 | 31.34 |
| Ours-Mistral-7B-instruct | 56.83 | 45.13 | 52.05 | 59.65 | 50.35 | 54.78 | 30.32 | 23.47 | 24.98 | 53.93 | 47.38 | 54.82 | 50.18 | 41.58 | 46.66 |
| Ours-Mistral-2501-24B | 59.89 | 47.62 | 56.39 | 67.03 | 54.51 | 59.98 | 35.84 | 23.68 | 28.54 | 60.81 | 53.19 | 61.59 | 55.89 | 44.75 | 51.63 |

**Baselines.** We compare EXSEARCH with a range of baselines, categorized into three groups based on their use of retrieval strategies: (i) **Direct reasoning without retrieval**: These methods produce the answer to the input query by prompting the LLM to reason over its parametric internal knowledge, without an external retriever. This includes few-shot prompting off-the-shelf LLMs, i.e., DeepSeek-R1 [17], GPT-4o [48], GPT-3.5 [47], Qwen2.5 [83], QwQ-32B [68], LLaMA-3.3-70B [13], and Mistral-8x7B [26]. (ii) **Advanced RAG**: These methods retrieve relevant documents, followed by filtering or re-ranking, and incorporate useful documents into the LLM's context for answer generation. We include: RankRAG [88], ChatQA [42], RetRobust [87], RAG-DDR [40] (trained by DPO [54]), and InstructRAG [74]. (iii) **Iterative RAG**: These methods allow LLMs to interact with the retriever iteratively. We include the widely-used methods: GenGround [64], DSPy [32], SearchChain [80], Iter-RetGen [61], Verify-and-Edit [94], Gen-Ret-Gen [1], Search-o1 [39], Search-R1 [28] (trained by DPO [60]), and Self-RAG [4]. We implement the above baselines following their official code. Full model descriptions are included in Appendix D.3. Following the most commonly used recipe for baselines, we set the size of retrieval documents to 10 for all advanced RAG baselines and 5 for all agentic RAG baselines (since they can iteratively retrieve), as well as for our method. Following RankRAG [88], for baselines without publicly available code, we report results from their original papers but only for reference (marked as "-" for metrics that were not reported in the original paper).

**Implementation details.** We apply the proposed method to various backbone LLMs, such as Qwen-2.5-7B-Instruct [83]. Following prior work [64, 80], we use the Wikipedia passage dump from December 20, 2018, as the retrieval corpus and adopt ColBERTv2.0 [59] for document retrieval. To

Table 2: Recall@K (K=3,5) for our method (*w/* 8B Llama3.1 and 7B Qwen2.5) and strong baselines.

| Tasks | NQ | | HotpotQA | | MusiQue | | 2WikiQA | |
|---|---|---|---|---|---|---|---|---|
| Metrics | R@3 | R@5 | R@3 | R@5 | R@3 | R@5 | R@3 | R@5 |
| ColBERTv2.0 | 70.32 | 77.64 | 46.20 | 51.79 | 14.27 | 17.91 | 29.13 | 32.36 |
| *Re-ranking* | | | | | | | | |
| MonoT5 [46] | 71.37 | 78.61 | 49.59 | 54.21 | 14.77 | 18.99 | 30.24 | 33.20 |
| BGE [76] | 78.47 | **81.86** | 52.83 | 55.89 | 16.80 | 20.23 | 32.83 | 34.44 |
| RankVicuna [51] | 69.36 | 76.66 | 49.28 | 52.86 | 15.56 | 18.08 | 29.66 | 32.38 |
| RankZepyhr [51] | 71.65 | 78.65 | 50.56 | 57.88 | 17.82 | 22.65 | 30.55 | 34.13 |
| *Query decomposition* | | | | | | | | |
| Search-o1 [39] | 69.76 | 71.76 | 45.71 | 65.66 | 20.37 | 30.28 | 32.81 | 36.22 |
| Search-r1 [28] | 68.21 | 70.00 | 42.71 | 65.41 | 18.45 | 25.48 | 27.73 | 34.52 |
| Ours-Qwen2.5-7B | 71.70 | 79.42 | **64.06** | 66.94 | **35.62** | 40.16 | **33.94** | 36.59 |
| Ours-Llama3.1-8B | 72.21 | 78.70 | 62.52 | 65.49 | 34.60 | **41.22** | 33.68 | 36.05 |

Table 3: Ablation study where we remove each component from the vanilla EXSEARCH.

| Tasks | HotpotQA | | | 2WikiQA | | |
|---|---|---|---|---|---|---|
| Metrics | F1 | EM | Acc. | F1 | EM | Acc. |
| Ours-Qwen2.5-7B | 62.59 | 50.35 | 54.32 | 57.14 | 52.62 | 54.37 |
| - *w/o thinking* | 54.23 | 43.46 | 51.45 | 45.68 | 42.36 | 48.35 |
| - *w/o search* | 43.34 | 39.61 | 42.25 | 42.15 | 39.01 | 41.46 |
| - *w/o recording* | 57.34 | 46.35 | 52.16 | 49.60 | 44.33 | 50.49 |
| - *w/o $w(z)$* | 57.23 | 43.36 | 47.45 | 53.35 | 46.55 | 49.35 |
| Ours-Llama3.1-8B | 60.72 | 47.59 | 53.59 | 54.62 | 47.48 | 54.21 |
| - *w/o thinking* | 52.31 | 41.31 | 45.21 | 48.42 | 42.13 | 47.87 |
| - *w/o search* | 48.25 | 39.24 | 41.86 | 41.24 | 35.32 | 39.34 |
| - *w/o recording* | 55.23 | 44.12 | 48.34 | 52.65 | 43.34 | 47.63 |
| - *w/o $w(z)$* | 54.45 | 43.56 | 49.23 | 49.61 | 42.51 | 49.54 |

equip the LLM with the basic skills required to follow our iterative *think-search-record* pattern, we initially generate 1,000 pseudo-examples and fine-tune the model on them, following a cold-start setup as in prior work [10, 66, 96]. Data collection details are provided in Appendix D.4. We also investigate the effect of data scale in the cold start on final performance (see § 5.4). For the subsequent EM training, we set the number of iterations $N$ to 5 and report model performance at each iteration. We use DeepSpeed ZeRO 3 [56] with a learning rate of $2 \times 10^{-6}$. More details are provided in § D.5.

# 5 Experimental Results

## 5.1 Overall evaluation

**Surpasses reasoning LLMs without retrieval.** As shown in Table 1, EXSEARCH substantially outperforms large-scale LLMs that rely solely on internal knowledge across all benchmarks. Compared with GPT-4o and LLaMA-3.3-70B (both using large-scale parameters), our method achieves improvements of +5.63 and +7.55 in average F1, respectively, despite using a much smaller 7B backbone model. These results highlight the limitations of closed-book reasoning and demonstrate the effectiveness of our interleaved search and reasoning approach in leveraging external knowledge.

**Outperforms RAG baselines.** EXSEARCH substantially outperforms strong RAG baselines, including both advanced RAG (e.g., RankRAG) and iterative variants (e.g., Search-o1). For example, on HotpotQA, it improves F1 from 55.40 (RankRAG-70B) and 53.31 (Search-o1-32B) to 62.59, using the 7B Qwen2.5 model. Even compared to Search-R1, trained by the cost-intensive PPO algorithm, our method achieves slightly higher performance (Avg. F1: 51.45 vs. 48.74) while using a much simpler self-improving loop based on the expectation-maximization framework. One reason for these improvements is that EXSEARCH enables the LLM to flexibly refine the query (*thinking*) and explicitly extract fine-grained evidence (*recording*), enabling more effective knowledge utilization.

**Improves retrieval performance.** To better understand the benefits of our agentic search method, we further evaluate the retrieval performance. We conduct experiments on four benchmarks and report *recall scores* in Table 2, which reflects how well the retrieved documents cover the ground truth. We follow prior work [31, 36] and define a retrieved document as correct if it contains the answer to the input question. The results reveal several key findings: (i) using the retriever alone yields low recall; (ii) conventional re-ranking methods like MonoT5 show relatively limited improvement in multi-hop QA scenarios; and (iii) our method, through iterative reasoning, achieves substantially higher recall. These results indicate that EXSEARCH, by expanding the search as reasoning exploration, can enhance retrieval performance in addition to improving end-to-end answer generation correctness.

## 5.2 Ablation studies

EXSEARCH consists of three core actions: *thinking*, *search*, and *recording*. To validate the effectiveness of each component, we independently remove each action and evaluate the resulting variant. Table 3 presents the results for the following:

(i) **w/o *thinking***: Removing *thinking* reduces our method to a standard retrieve-then-generate pipeline, leading to a substantial performance drop across all datasets. This validates the importance of reasoning and iterative query refinement in enabling more effective retrieval.

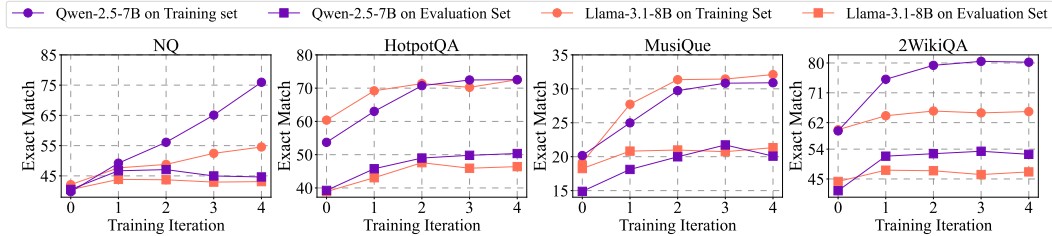

Figure 3: Training convergence of Qwen-2.5-7B and Llama-3.1-8B, where we report the *Exact Match* score for checkpoints in each iteration.

  (ii) **w/o *search***: Without *search*, the method becomes equivalent to an iterative self-ask process without external knowledge. We observe a consistent decline across all datasets, validating the necessity of integrating retrieval into the reasoning process.

 (iii) **w/o *recording***: Disabling *recording* prevents the model from extracting key evidence for each sub-query, resulting in an average performance drop of approximately 8%, as shown in Table 3. This suggests that deliberately and explicitly recording fine-grained evidence can enhance the model's understanding of external documents, which is consistent with prior findings in interpretability-focused studies, such as Physical LLMs [3].

 (iv) **w/o $w(z)$**: Removing the weighting function $w(z)$ reduces our method to a naive self-training setup, where the model is directly trained on its generated data. Results in Table 3 show a substantial performance drop, e.g., the F1 score on the HotpotQA dataset drops from 60.72 to 54.45. This highlights the critical role of $w(z)$ in guiding the model to prioritize trajectories that contribute to correct answers, rather than treating all sampled actions equally.

## 5.3 Training convergence

The theoretical analysis in § 3 demonstrates the non-decreasing improvement property during our training process. To empirically verify this property, we evaluate model checkpoints after each training iteration on four experimental benchmarks. Figure 3 shows the results, with more details provided in Appendix D.6. We observe that the performance on the training set consistently improves over iterations and typically stabilizes, validating the expected non-decreasing convergence behavior. These findings align well with our theoretical expectations and further demonstrate the practical efficiency of our optimization strategy. On the evaluation sets, the model typically reaches peak performance within the second or third iteration, indicating rapid convergence in practice.

## 5.4 Investigating the cold start: Is more warm up necessary?

In our main experiments (Table 1), EXSEARCH is initialized with 1,000 synthetic examples for warm-up empirically, following prior work [10, 96]. To explore the necessity and impact of warm-up data, we train separate models with varying amounts of supervised fine-tuning (SFT) data (denoted as $K$) and apply EXSEARCH to each model independently. As shown in Figure 4, we observe three key trends: (i) **Warm-up helps**: Initial training equips the model with basic reasoning capabilities and improves final performance; (ii) **More data is not always necessary**: Qwen2.5-7B, even without any warm-up (*zero*), already outperforms strong baselines; (iii) **100 examples offer a good trade-off**: Using 100 SFT examples strikes a balance between performance and the cost of data synthesis (approximately 2$). We provide additional results and analysis in Appendix D.6.

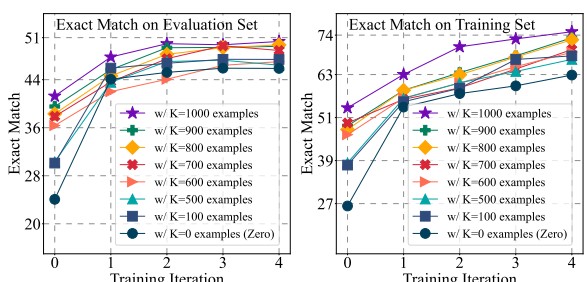

Figure 4: Performance for Qwen2.5-7B that is initially empowered by different amounts of warm-up data.

## 5.5 Case studies

To gain deeper insights, we conduct qualitative case studies using Qwen2.5-7B, with concrete cases provided in Appendix F.3. We observe two key strengths: (i) The model integrates reasoning and retrieval steps, dynamically selecting relevant evidence as part of its reasoning process; (ii) By interleaving reasoning with retrieval, our method helps the model ground its generated answers in factually relevant knowledge. We also manually examine 200 random failure cases and identify the following two errors: **Under-searching**: In 3.5% of cases, the model is misled by plausible-looking but incomplete evidence, retrieves fewer documents than needed, and stops too early; and **Over-searching**: In 7.5% of cases, the model retrieves more documents than necessary but still misses key information, failing to recognize when to stop. These observations highlight the need for better search-depth control and suggest improving trajectory-level stopping criteria in future work.

## 6 EXSEARCH-Zoo: An Extended Suite

So far, we have validated the effectiveness of EXSEARCH. The promising results in § 5 motivate us to extend our method to more scenarios. We therefore introduce EXSEARCH-Zoo, a resource extending our method in two dimensions: *diverse backbone models* and *extendable retrieval strategy*.

**Diverse backbone models.** We apply EXSEARCH to a wide range of LLMs and analyze the generalizability across LLMs of varying sizes and model families. Table 1 and Figure 1 show the experimental results, which exhibit a clear scaling-law pattern. In more details, we highlight two key observations: (i) there is a consistent performance improvement as the model size increases; and (ii) even models as small as 3B parameters achieve competitive performance when trained by our method. These suggest that EXSEARCH is broadly applicable and scales across different models.

**Extended retrieval strategy.** In EXSEARCH, the LLM explores a search trajectory by iteratively thinking, searching, and recording. However, to answer more complex queries, we may need to customize a more specific retrieval strategy beyond these three actions. To demonstrate the extensibility, we introduce *an example by adding an additional document selection* action, extending the retrieval process into the *think → search → select → record* pattern. Document selection allows the model to prioritize relevant evidence and discard distracting content, a technique widely used in the IR field. Specifically, the LLM reads the retrieved documents and autoregressively generates a ranked list of selected

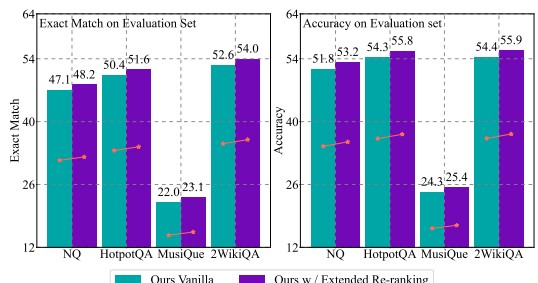

Figure 5: Performance of EXSEARCH when extended with the document re-ranking action.

identifiers (e.g., [1] > [2] > [3]), similar to **list-wise re-ranking** [51, 67]. This variant modifies only the E-step by adding the selection action to the trajectory, while the overall EM training remains a self-improving process. Figure 5 shows consistent improvements, with accuracy gains of +1.1 and +1.5 on MuSiQue and 2WikiQA, respectively. These results suggest that more fine-grained or customized actions can be seamlessly incorporated into EXSEARCH. A detailed derivation and experimental results are provided in Appendix 6.

## 7 Related Work

**Information retrieval with LLMs.** As one of the most fundamental data mining techniques, information retrieval (IR) primarily focuses on understanding input queries, extracting relevant information from external data sources, and providing accurate responses to users [18, 34]. In recent years, large language models (LLMs), pre-trained on large-scale corpora, have been integrated into IR systems to enhance conventional retrieval methods [15, 77, 81], leveraging their language processing abilities, such as LLM-based query rewriting. More recent work also uses LLMs to aggregate long-context retrieved information and generate concise answers for users, a process known as retrieval-augmented generation (RAG) [15]. However, most existing approaches integrate LLMs into IR by

cascading traditional retrieval pipelines and optimizing LLMs for specific stages. For example, Self-RAG [4] fine-tune the LLMs by document relevance judgment and answer generation; ADEIE [52] fine-tunes LLMs for knowledge extraction; and other studies align LLMs for improved query decomposition [28, 40]. In contrast, we model the *entire retrieval* process through a dynamic and iterative agentic framework, where LLMs are self-incentivized using the expectation-maximization algorithm. *We also discuss further distinctions in Appendix B.*

**Reasoning by LLMs.** LLMs have demonstrated strong capabilities in long-term planning [8, 86], often decomposing complex tasks into a sequence of sub-tasks and solving them step by step using their internal parametric knowledge [73]. This ability, typically referred to as *reasoning* [50, 53, 79] or *deliberate thinking* [24, 29, 85], underpins many recent advancements in LLM performance. For example, models such as GPT-4o [48] and DeepSeek-R1 [17], which are post-trained using reinforcement learning (e.g., PPO [60] or GRPO [9, 62]), have achieved impressive results on structured tasks like mathematical reasoning [41] and code generation [27, 95]. However, LLMs suffer from outdated internal knowledge and the tendency to hallucinate factually incorrect content. Such closed-book reasoning often underperforms in knowledge-intensive tasks like multi-hop question answering, where models need to access and integrate factual information [16, 39]. In this work, we tightly integrate retrieval with the LLM reasoning process. Unlike prior methods that directly augment LLMs with documents relevant to input queries [55, 74], our approach allows the model to acquire external evidence dynamically as reasoning unfolds and learns this pattern autonomously.

## 8    Conclusion and Future Work

In this work, we propose EXSEARCH, a novel framework that empowers an LLM to become an exploratory search agent, capable of iteratively reasoning over context, searching for target information, and reflecting on retrieved documents. EXSEARCH incorporates a self-incentivized loop, where the LLM learns to reason and search from its own generated data, progressively enhancing its expertise as an effective search agent. We provide a theoretical analysis of the advantages of EXSEARCH and conduct experiments on a wide range of datasets to demonstrate its effectiveness. Motivated by its strong performance, we introduce EXSEARCH-Zoo, a comprehensive resource that extends our vanilla method in two directions: (i) backbone LLM diversity and (ii) enriched action spaces (e.g., document re-ranking). Further experiments also validate the extensibility of our framework. We suggest future work to (i) extend EXSEARCH beyond text-only reasoning to include multi-modal reasoning with vision inputs, and (ii) apply EXSEARCH to tool-augmented agents, thereby supporting more real-world tasks.

## Acknowledgements

This research was (partially) supported by the Dutch Research Council (NWO), under project numbers 024.004.022, NWA.1389.20.183, and KICH3.LTP.20.006, and the European Union under grant agreements No. 101070212 (FINDHR) and No. 101201510 (UNITE).

Views and opinions expressed are those of the authors only and do not necessarily reflect those of their respective employers, funders and/or granting authorities.

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

# Appendix

## Contents

## A  Practical Considerations and Societal Impact

### A.1  Limitations

Despite our efforts to improve the LLM's ability to reason and search, our work has several limitations. Some of these limitations are shared across many existing RAG systems. We highlight them not only as directions for future work but also in the hope of inspiring broader exploration within the community to advance the development of more effective retrieval-augmented reasoning systems.

First, our method currently focuses on textual inputs and outputs. Extending reasoning-augmented search to multimodal scenarios, e.g., incorporating images or structured tables, remains an important direction for future work. We plan to integrate EXSEARCH with large vision-language models and extend the text-only retrieval documents to a multimodal corpus.

Second, in line with prior work, we deliberately avoid hardcoding heuristics such as fixed query decomposition rules or predefined in-context learning demonstrations. However, similar to most existing approaches, our model performs retrieval at every reasoning step, regardless of necessity. Since LLMs pre-trained on large web data have parameterized extensive world knowledge, for some

simpler queries, this fixed retrieval strategy may be redundant. An important next step is to develop more adaptive strategies that allow the model to decide *when* to retrieve based on context, rather than always retrieving at each step.

Third, our training and evaluation are based on rule-based, answer-centric metrics such as Exact Match and F1 score. While effective for benchmark evaluation, these metrics may not fully capture performance in more open-ended or exploratory tasks, especially for long-form generation. As mentioned in our paper, in future work, we aim to explore more open-domain setups and alternative supervision signals beyond gold-standard answers, such as LLM-as-the-judge.

Overall, the proposed EXSEARCH makes progress in the RAG research area by effectively teaching LLMs to interleave reasoning and search within a self-improving process. However, it remains an initial step. We believe addressing the above limitations will be crucial for building more general and robust search-augmented reasoning systems.

## A.2  Ethics statement

The research conducted in this paper centers around the development of a reasoning-augmented search framework. The proposed method enables language models to dynamically retrieve and reason over external information. In the process of conducting this research, we have adhered to ethical standards to ensure the integrity and validity of our work. All the questions used in this study were obtained from existing benchmarks, ensuring a high level of transparency and reproducibility in our experimental procedure. To support our retrieval system, we used an open-source corpus, specifically Wikipedia. This ensures that our research utilizes publicly accessible and freely available data, minimizing potential bias and promoting fairness.

We have made every effort to ensure that our study does not involve human subjects, private data, or any content that may cause harm to individuals or social groups. No part of this work includes deceptive practices or intentional misuse of information. We are committed to conducting and presenting this research with integrity and social responsibility. We intend to release our code and implementation details to support open research, following the NeurIPS submission policy, and aim to facilitate further study in the information retrieval and retrieval-augmented generation areas.

## A.3  Societal impacts

Our work introduces EXSEARCH, a reasoning-augmented search framework that interleaves multi-step reasoning with dynamic document retrieval. The primary goal is to improve the factual accuracy and interpretability of LLMs in open-domain question answering and knowledge-intensive tasks. The model presented in this work requires explicit grounding in external sources, and our framework emphasizes verifiable retrieval and intermediate reasoning steps, which improve transparency. These properties support downstream mitigation efforts, such as traceable generation or retrieval auditing. Additionally, we note that our current implementation does not involve user data or personalized profiles, thus mitigating privacy concerns.

This approach offers several key benefits. *Improved Transparency*: By grounding answers in external sources and providing intermediate reasoning steps, EXSEARCH allows for better traceability of the model's decision-making process, which is crucial for understanding and verifying outputs; *Enhanced Trustworthiness*: The framework's reliance on verifiable external retrieval reduces the risk of generating hallucinated information, contributing to more reliable and factually accurate responses; and *Broader Applicability*: With its focus on factual grounding and reasoning, EXSEARCH can be applied to a wide range of knowledge-intensive applications, including scientific research, legal analysis, and education, where accuracy and clarity are essential.

While this research is foundational and not tied to specific applications or deployments, we acknowledge the broader risks associated with enhancing the factual accuracy and coherence of LLMs. Specifically, the improved text generation capabilities of LLMs may be misused in malicious contexts or scenarios, such as: (i) generating more convincing disinformation; (ii) fabricating plausible but incorrect content by selectively retrieving or combining real documents; and (iii) enabling better-targeted persuasive text (e.g., phishing, political propaganda). In summary, while EXSEARCH could be misused in unintended settings, its design principles, combined with the potential for verifiable and auditable generation, offer avenues for responsible deployment. While we do not foresee imme-

diate or direct societal harm, we encourage future work to explore safeguards, such as automated monitoring and ethical auditing mechanisms, in high-stakes applications.

# B Comparison with Prior Work

In this work, the proposed method enables LLMs to perform interleaved reasoning and search (also referred to as agentic search in this work) and optimizes their capability for this pattern through a self-incentivized framework. Below, we compare our method with previous iterative retrieval methods and self-learning methods in detail.

## B.1 Comparing with previous iterative retrieval method

In this paragraph, we systematically compare EXSEARCH with previous work that incorporates iterative retrieval techniques, especially those integrated with LLMs, highlighting our fundamental innovations. Most previous iterative retrieval approaches rely on a modular multi-stage pipeline, where distinct models are trained separately for sub-tasks such as query rewriting, retrieval, and evidence aggregation. For example, For example, Self-RAG [4] and RankRAG [88] train LLMs for document relevance judgment; ADELIE [52] adopts LLMs for knowledge extraction; while other work [2, 43] cascades multiple specialized components in sequence. Despite their progress, this modular design overlooks the end-to-end optimization, leading to potential misalignment among the training objectives in different stages. More recently, several works attempt to prompt powerful, extremely large models (i.e., GPT-3.5 or Qwen-32B) to iteratively interact with retrieval engines through in-context learning [39, 61]. However, they are limited by using predefined demonstrations or in-context learning examples. While simplifying system design, these approaches overlook improving the LLM's intrinsic reasoning capability for adaptive retrieval and generation. In contrast, EXSEARCH trains a *unified* LLM to reason over the evolving context, adaptively decide what information to retrieve, optionally re-rank retrieved documents, extract fine-grained supporting evidence, and generate the final answer, all within an end-to-end learning framework. Unlike previous work, EXSEARCH aligns the training signals of these actions through a coherent learning objective, improving the LLM in an end-to-end manner.

*The most contemporary work to* EXSEARCH *is Search-R1 [28], developed independently around the same time*. Search-R1 applies the proximal policy optimization (PPO [60]) algorithm to encourage LLMs to issue multiple search queries during reasoning. However, EXSEARCH differs from Search-R1 in three critical aspects: (i) *Reasoning Process*: EXSEARCH enables richer retrieval actions, including query generation, optional re-ranking of retrieved documents, and fine-grained evidence extraction, while Search-R1 only conducts iterative query decomposition without reflection or re-ranking; (ii) *Training Algorithm*: EXSEARCH treats search trajectories as latent variables and optimizes a variational evidence lower bound via a Generalized Expectation-Maximization algorithm [45], achieving stable and progressive self-improvement. In contrast, Search-R1 adopts an online on-policy reinforcement learning framework, which often suffers from sample inefficiency and training instability [19, 21, 23]; and (iii) *Reward Signal*: EXSEARCH introduces a trajectory-level training signal, evaluating the quality of the entire search trajectory based on the likelihood of generating the correct answer, rather than relying solely on a binary outcome-based reward as in Search-R1.

## B.2 Comparison with previous self-training methods

Recent studies have explored self-training frameworks where a model is iteratively trained on its generated data [7, 75]. For example, some work [7, 65] allows the model to first generate a solution and fine-tune it on the generated trajectories, showing promising results on mathematical reasoning tasks. Similar ideas have also been applied in combination with REINFORCE Leave-One-Out (RLOO) methods [7, 35, 91]. More recently, other work [11, 75] proposes self-rewarding methods, where the LLM itself is used via LLM-as-a-Judge prompting to provide its own rewards during an iterative DPO [54] training process. While effective in closed-book settings, these approaches overlook a critical limitation: **the inherent limitation of parametric knowledge in LLMs** [15, 16]. Without external retrieval, the model cannot dynamically access relevant information when it is missing from its internal memory. In contrast, this work focuses on agentic search, which aims to enable the LLM to interleave dynamic retrieval within the reasoning process and further reflect on the

Table 4: Main notation used in this work.

| Symbol | Description |
|---|---|
| $x$ | The initial input query. |
| $y$ | The ground-truth answer corresponding to the initial query $x$. |
| $\theta$ | The parameters of the large language model (LLM). |
| $\mathcal{R}$ | The external retriever. |
| $i$ | The index of the $i$-th step in the reasoning and retrieval process. |
| $x_i$ | The sub-query generated at the $i$-th step. |
| $\boldsymbol{d}_i$ | The set of documents retrieved in the $i$-th step, i.e., $\boldsymbol{d}_i = \mathcal{R}(x_i) = \{\boldsymbol{d}_{i,j} \mid j \in [K]\}$. |
| $e_i$ | The fine-grained evidence extracted from the retrieved documents $\boldsymbol{d}_i$. |
| $t$ | The index of the $t$-th training iteration. |
| $\boldsymbol{z}$ | The full reasoning trajectory, consisting of interleaved sub-queries, retrieved documents, and extracted evidence, i.e., $\boldsymbol{z} = \{(x_i, \boldsymbol{d}_i, e_i) \mid i \in [|\boldsymbol{z}|]\}$ |

retrieved content at a fine-grained level (See the § 2.1 in the main body of the paper). Additionally, we provide a theoretically grounded analysis of the advantages of our method, as discussed in this Appendix C.2 and Appendix C.3. We also introduce an extended resource, EXSEARCH-Zoo, which supports multiple model families and richer reasoning actions.

# C  Detailed Theoretical Analysis

Due to space constraints in the main body of this paper, we include the detailed version of the theoretical derivations and analyses of EXSEARCH here. Below, we first formulate the search and reasoning process in EXSEARCH. We then introduce how the generalized expectation–maximization technique is leveraged to improve the LLM's capability in EXSEARCH through a self-improving loop (§ C.1). Additionally, we prove that the resulting training procedure is convergent. Table 4 lists the main notation used in the paper.

## C.1  Skeleton derivations for EXSEARCH

**Reviewing agentic search procedure.**  In this work, the proposed EXSEARCH is inspired by the *Exploratory Search* paradigm [44], which models information-seeking as a dynamically unfolding process where search queries are iteratively refined based on intermediate results. EXSEARCH simulates this by interleaving three core actions:

(i) *thinking*: The LLM generates a query $x_i$ based on the current context $x$ and the accumulated search trajectory $\boldsymbol{z}_{<i}$, formulated as:

$$x_i = p(x_i \mid x, \boldsymbol{z}_{<i}; \theta) \tag{10}$$

(ii) *search*: A retrieval module $\mathcal{R}$ retrieves the top-$K$ documents $\boldsymbol{d}_i$ relevant to the query $x_i$:

$$\boldsymbol{d}_i = \mathcal{R}(x_i) \tag{11}$$

(iii) *recording*: The LLM reflects on the retrieved documents $\boldsymbol{d}_i$ and extracts evidence $e_i$ conditioned on $x$ and $\boldsymbol{d}_i$:

$$e_i = p(e_i \mid x, \boldsymbol{d}_i; \theta) \tag{12}$$

In this step, the model focuses solely on the current sub-query and its associated documents, reducing computational cost by limiting the context to the most relevant information.

Formally, the reasoning-augmented search process is modeled as a sequence $\boldsymbol{z} = \{(x_i, \boldsymbol{d}_i, e_i) \mid i \in [|\boldsymbol{z}|]\}$, with the joint likelihood:

$$p(\boldsymbol{z} \mid x; \theta) = \prod_{i=1}^{|\boldsymbol{z}|} p\left((x_i, \boldsymbol{d}_i, e_i) \mid x, \boldsymbol{z}_{<i}; \theta\right) \tag{13}$$

After the interleaved search and reasoning process, the LLM aggregates information from $\boldsymbol{z}$ to generate the final answer $y \sim p(y \mid x, \boldsymbol{z}; \theta)$.

**Training objective.** The goal of EXSEARCH is to improve the LLM's ability to generate the correct answer $y$ after reasoning. The training objective is formulated as:

$$\log p(y \mid x; \theta) = \log \sum_{\boldsymbol{z}} p(y, \boldsymbol{z} \mid x; \theta) \tag{14}$$

Here, $\boldsymbol{z} = \{(x_i, \boldsymbol{d}_i, e_i) \mid i \in [\|\boldsymbol{z}\|]\}$ represents a sequence of the *thinking*, *search*, and *recording* actions. In EXSEARCH, we introduce a proposal distribution $q(\boldsymbol{z} \mid x)$ to approximate the sampling space of $\boldsymbol{z}$ and apply Jensen's inequality to the marginal log-likelihood in Eq. 14:

$$
\begin{aligned}
\log \sum_{\boldsymbol{z}} q(\boldsymbol{z} \mid x) &\frac{p(y, \boldsymbol{z} \mid x; \theta)}{q(\boldsymbol{z} \mid x)} \\
&\geq \sum_{\boldsymbol{z}} q(\boldsymbol{z} \mid x) \log \frac{p(y, \boldsymbol{z} \mid x; \theta)}{q(\boldsymbol{z} \mid x)} \\
&= \mathbb{E}_{\boldsymbol{z} \sim q(\boldsymbol{z}|x)} \left[ \log \frac{p(y, \boldsymbol{z} \mid x; \theta)}{q(\boldsymbol{z} \mid x)} \right]
\end{aligned}
\tag{15}
$$

The right-hand side represents the variational evidence lower bound (ELBO) of $\log p(y \mid x; \theta)$, and the bound becomes tight if $q(\boldsymbol{z} \mid x)$ approximates the true posterior distribution $p(\boldsymbol{z} \mid y, x; \theta)$. Thus, we can iteratively estimate $p(\boldsymbol{z} \mid y, x; \theta)$ and maximize this ELBO using the generalized expectation-maximization algorithm to progressively improve the LLM.

**E-step: Trajectory exploration.** In the E-step, we estimate the distribution over reasoning trajectories $\boldsymbol{z}$ by sampling from the LLM $\theta$. In the $t$-th iteration, we approximate the distribution $q(\boldsymbol{z} \mid x) \approx p(\boldsymbol{z} \mid x, y; \theta)$, yielding the following form for the ELBO:

$$\text{ELBO} = \mathbb{E}_{\boldsymbol{z} \sim p(\boldsymbol{z}|x,y;\theta^t)} \left[ \log p(y, \boldsymbol{z} \mid x; \theta) \right] + \mathcal{H}(p(\boldsymbol{z} \mid x, y; \theta^t)) \tag{16}$$

Here, the entropy term $\mathcal{H}(p(\boldsymbol{z} \mid x, y; \theta^t))$ is constant with respect to $\theta$. Since direct sampling from the posterior is intractable, we apply importance sampling [14, 69], where the distribution $p(\boldsymbol{z} \mid x; \theta^t)$ is easier to sample from, and each sample is assigned an importance weight:

$$w(\boldsymbol{z}) = \frac{p(\boldsymbol{z} \mid x, y; \theta^t)}{p(\boldsymbol{z} \mid x; \theta^t)} \tag{17}$$

This allows us to rewrite the ELBO as:

$$\text{ELBO} = \mathbb{E}_{\boldsymbol{z} \sim p(\boldsymbol{z}|x;\theta^t)} \left[ w(\boldsymbol{z}) \log p(y, \boldsymbol{z} \mid x; \theta) \right] + c, \tag{18}$$

where $c$ is a constant.

**M-step: Re-weighted trajectory learning.** In the M-step, we update the model parameters $\theta$ by maximizing the ELBO from Eq. 18. The objective becomes:

$$\theta = \arg\max_{\theta} \mathbb{E}_{\boldsymbol{z} \sim p(\boldsymbol{z}|x;\theta^t)} \left[ w(\boldsymbol{z}) \log p(y, \boldsymbol{z} \mid x; \theta) \right] \tag{19}$$

The overall training process is performed using stochastic gradient descent, with gradients computed as:

$$\nabla_{\theta} \text{ELBO}(\theta) = -\mathbb{E}_{\boldsymbol{z} \sim p(\boldsymbol{z}|x;\theta^i)} \left[ w(\boldsymbol{z}) \nabla_{\theta} (\mathcal{L}_{\mathcal{R}} + \mathcal{L}_{\mathcal{A}}) \right] \tag{20}$$

### C.2 Convergence analysis

Below, we analyze the convergence behavior of EXSEARCH using the generalized expectation-maximization algorithm, providing a more detailed explanation than in the main body of the paper. We show that the training objective $\log p(y \mid x; \theta)$ is non-decreasing after each training iteration and progressively converges to a stationary point due to its upper-bounded property. To provide a tighter characterization of convergence, we interpret the optimization gap as a KL divergence between importance-weighted sampling and the true posterior.

**Lemma C.1** (Monotonic improvement). *At each iteration $t \in \mathbb{Z}^+$, the training objective of the LLM satisfies:*

$$\log p(y \mid x; \theta^{t+1}) \geq \log p(y \mid x; \theta^t). \tag{21}$$

*Proof.* Reviewing the EM-style training in our method, the main concept involves introducing a tractable evidence lower bound (ELBO) and progressively improving it to optimize $\log p(y|x;\theta)$. In the E-step of the $t$-th iteration, we sample trajectories $\boldsymbol{z}$ from the current model as $\boldsymbol{z} \sim p(\boldsymbol{z} \mid x;\theta^t)$ and assign each a weight $w(\boldsymbol{z}) = \frac{p(\boldsymbol{z}|x,y;\theta^t)}{p(\boldsymbol{z}|x;\theta^t)} \propto p(y \mid x, \boldsymbol{z};\theta^t)$. We define the ELBO as:

$$\text{ELBO}(\theta, \theta^t) := \mathbb{E}_{\boldsymbol{z} \sim p(\boldsymbol{z}|x;\theta^t)} [w(\boldsymbol{z}) \log p(y, \boldsymbol{z} \mid x;\theta)] + c. \tag{22}$$

In the M-step, we update the model by maximizing this ELBO:

$$\theta^{t+1} = \arg\max_\theta \text{ELBO}(\theta, \theta^t), \tag{23}$$

which guarantees that: $\text{ELBO}(\theta^{t+1}, \theta^t) \geq \text{ELBO}(\theta^t, \theta^t)$. Meanwhile, since the ELBO indicates the evidence lower bound for the marginal distribution $\log p(y \mid x;\theta^{t+1})$, it holds that: $\log p(y \mid x;\theta^{t+1}) \geq \text{ELBO}(\theta^{t+1}, \theta^t)$, and $\text{ELBO}(\theta^t, \theta^t) = \log p(y \mid x;\theta^t)$. By combining these two equations, we have:

$$\log p(y \mid x;\theta^{t+1}) \geq \log p(y \mid x;\theta^t). \tag{24}$$

$\square$

Therefore, we have completed the proof of non-decreasing improvement for each training iteration.

**Lemma C.2** (Boundedness). *The sequence $\{\log p(y \mid x;\theta^t)\}_{t=1}^\infty$ is upper-bounded.*

*Proof.* Since $p(y \mid x;\theta) \in [0, 1]$, we naturally have $\log p(y \mid x;\theta) \leq 0$ for all $\theta$. $\square$

**Theorem C.3** (Convergence of EXSEARCH). *By Lemma C.1 and Lemma C.2, the sequence $\{\log p(y \mid x;\theta^t)\}$ is non-decreasing and upper-bounded. By the Monotone Convergence Theorem [5], it converges to a finite limit.*

**Remark C.1** (Tightness via KL Divergence). *The ELBO can be interpreted as a tight bound of $\log p(y \mid x;\theta)$ with the following identity:*

$$\log p(y \mid x;\theta) = ELBO(\theta, \theta^t) + KL(q^*(\boldsymbol{z}) \parallel p(\boldsymbol{z} \mid x, y;\theta)), \tag{25}$$

*where $q^*(\boldsymbol{z}) \propto w(\boldsymbol{z}) \cdot p(\boldsymbol{z} \mid x;\theta^t)$ is the induced sampling distribution. Therefore, under mild regularity conditions (e.g., bounded support, continuity of log-likelihood), as $q^*$ approaches the true posterior, the KL term vanishes, and the ELBO becomes tight. This strengthens the convergence result and characterizes the optimization gap.*

## C.3 Rethinking what the LLM learns within EXSEARCH

In § C.1 and main body of our paper, we demonstrate that the answer log-likelihood (i.e., $p(y|x, \boldsymbol{z};\theta)$) serves as an end-to-end training signal in our self-incentivized training process, guiding the model to reason and search. Given the primary goal of information retrieval, especially in downstream tasks like retrieval-augmented generation, we typically train the model to generate accurate answers for users and evaluate its performance using correctness-oriented metrics, such as exact match score or accuracy, thereby ensuring the factuality. However, a natural question arises:

> *Does our self-improving framework also maximize the commonly used downstream metrics, such as accuracy, and why does it work or not?*

In addition to the strong empirical results shown in our experiments, we provide a more interpretable and theoretical analysis in this section. Below, we first briefly highlight the learning objective introduced in § C.1, denoted as the **vanilla objective**, which uses $w(\boldsymbol{z}) \propto p(y \mid x, \boldsymbol{z};\theta)$ as the training signal. We then introduce a **goal-oriented learning objective**, where the LLM is trained to maximize an evaluation metric using a similar Expectation-Maximization algorithm. Finally, we relate these two objectives and show their consistency in model training and optimization, illustrating why the vanilla objective aligns with maximizing the expected metric and downstream task performance.

### C.3.1 Reviewing vanilla learning objective

The vanilla learning objective is defined as $\text{ELBO} = \mathbb{E}_{\boldsymbol{z} \sim p(\boldsymbol{z}|x;\theta^t)} [w(\boldsymbol{z}) \log p(y, \boldsymbol{z} \mid x;\theta)]$. Here, $w(\boldsymbol{z})$ is the weighting function given by $w(\boldsymbol{z}) = \frac{p(\boldsymbol{z}|x,y;\theta^t)}{p(\boldsymbol{z}|x;\theta^t)}$. This weighting function is derived from the ratio of the posterior distribution $p(\boldsymbol{z} \mid x, y;\theta^t)$ to the prior $p(\boldsymbol{z} \mid x;\theta^t)$, which reflects how well the trajectory $\boldsymbol{z}$ supports the final answer $y$.

### C.3.2 Introducing goal-oriented objective

We now replace the weighting function $w(z)$ with a evaluation metric $r(y)$ (*also widely known as the reward function*), which evaluates the quality of the final answer $y$. Formally, given an evaluation metric $r(y)$ that rates the quality of an answer $y$, our goal is to optimize the model parameters $\theta$ to improve the expected performance. We define the expected learning objective under the model as:

$$\mathcal{J}(\theta) = \mathbb{E}_{y \sim p(y|x;\theta)}[r(y)] = \sum_y r(y) p(y \mid x) \tag{26}$$

In EXSEARCH, since the model first generates a reasoning trajectory $z$ and then outputs an answer $y$, we can rewrite Eq. 26 as:

$$\mathcal{J}(\theta) = \mathbb{E}_{y \sim p(y|x;\theta)}[r(y)] = \sum_y r(y) \sum_z p(z, y \mid x) = \sum_{z,y} r(y) p(y, z \mid x). \tag{27}$$

Here, $p(y, z \mid x)$ denotes the LLM generating a reasoning path $z$ followed by a final answer $y$. Marginalizing over all possible $(z, y)$ is typically intractable due to the large action space of the LLM. We now derive a variational surrogate for optimizing such a goal-oriented objective through a tractable lower bound.

**Proposition 1** (Variational lower bound as a proxy for metric maximization). *Given a non-negative evaluation metric function $r$, let $\mathcal{J}(\theta) := \mathbb{E}_{z,y \sim p(z,y|x;\theta)}[r(y)]$ be the expected metric. We can introduce a proposal distribution $q(z, y)$ over the $(z, y)$ space to construct a more tractable evidence lower bound:*

$$ELBO(\theta, q) := \sum_{z,y} q(z, y \mid x) \log \frac{r(y) p(z, y \mid x; \theta)}{q(z, y \mid x)}, \tag{28}$$

*which is a function only related to $q$ and $\theta$. It satisfies:*

$$\left\| \arg\max_\theta ELBO(\theta, q) - \arg\max_\theta \mathcal{J}(\theta) \right\| \leq c \cdot (KL(q \parallel r \cdot p_\theta))^{1/2}. \tag{29}$$

*, where $c$ is a constant. Under mild assumptions, the boundedness (=) holds when $q = q^* \approx r \cdot p_\theta$.*

We provide the detailed proof for this proposition in § C.3.5. This proposition indicates that we can optimize the ELBO as a proxy to improve the $\mathcal{J}(\theta)$, following a similar Expectation-Maximization algorithm as introduced in vanilla EXSEARCH (§ C.1). In more details, this involves alternating between the following steps: (i) E-step: sampling $(z, y)$ trajectories; and (ii) M-step: updating the model parameters.

**E-step: Sampling trajectories.** To approximate the true posterior distribution $q^*$, we sample $(z, y)$ from the current model $p(z, y \mid x; \theta^t)$ using an importance sampling strategy. The corresponding importance weight is formulated as $\frac{q^*(z,y|x)}{p(z,y|x;\theta^t)} = r(y)$, which is obtained using the evaluation metric.

$$\begin{aligned} \text{ELBO} &= \mathbb{E}_{(z,y) \sim q^*(z,y|x)}[\log p(z, y \mid x; \theta)] + c \\ &= \mathbb{E}_{(z,y) \sim p(z,y|x;\theta^t)} \left[ \frac{q^*(z,y)}{p(z,y \mid x;\theta^t)} \log p(z, y \mid x; \theta) \right] + c \\ &= \mathbb{E}_{(z,y) \sim p(z,y|x;\theta^t)}[r(y) \log p(z, y \mid x; \theta)] + c \end{aligned} \tag{30}$$

**M-step: Update the model parameters.** Given the weighted samples, we update $\theta$ by maximizing the weighted log-likelihood:

$$\theta = \arg\max_\theta \mathbb{E}_{(z,y) \sim p(\cdot|\theta^t)}[r(y) \log p(z, y \mid x; \theta)]. \tag{31}$$

By alternating between the above E-step and M-step, we can train the LLM to maximize the given metric. This can be seen as a reward-weighted generalization of expectation-maximization [12]. This formulation naturally integrates downstream evaluation metric (via $r(y)$) into a likelihood-based training framework.

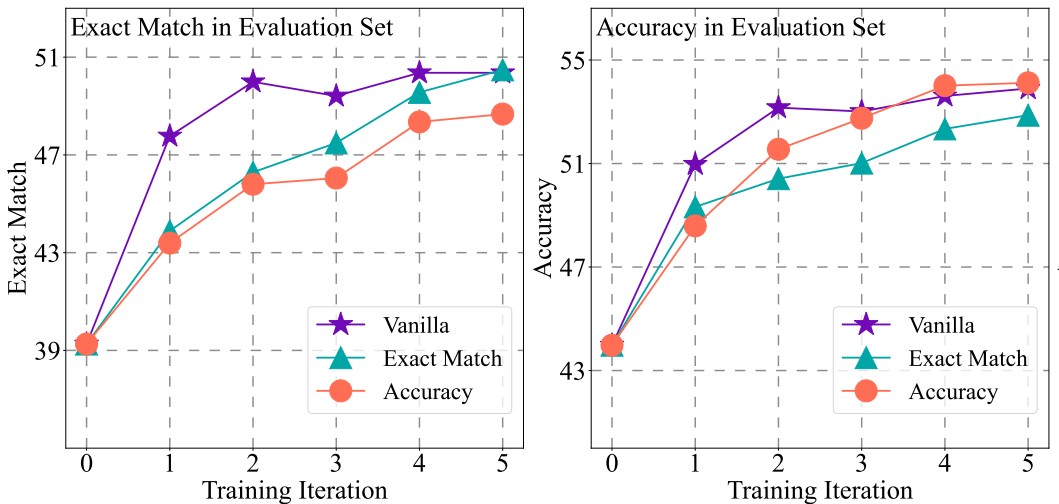

Figure 6: Comparing vanilla training process in our method with variants trained using Exact Match or Accuracy as the training signal.

### C.3.3 Relating vanilla learning objective with goal-oriented objective

In the goal-oriented objective, the evaluation metric $r(y)$ directly evaluates the quality of the answer $y$ to return a training signal. Similarly, the training signal $w(\boldsymbol{z}) \propto p(y \mid x, \boldsymbol{z}; \theta)$ represents the probability of generating a correct answer, and the higher the probability of generating the correct answer, the greater the expected metric. Therefore, we have $w(\boldsymbol{z}) \propto r(y)$.

Reviewing the optimization objectives in the *vanilla objective* and the *goal-oriented objective*: In the vanilla objective, we have $\theta = \arg\max_\theta \mathbb{E}_{\boldsymbol{z} \sim p(\boldsymbol{z}|x;\theta^t)} [w(\boldsymbol{z}) \log p(y, \boldsymbol{z} \mid x; \theta)]$. In the goal-oriented objective, we have $\hat{\theta} = \arg\max_\theta \mathcal{J}(\theta) \approx \arg\max_\theta \mathbb{E}_{(\boldsymbol{z},y) \sim p(\cdot|\theta^t)}[r(y) \log p(\boldsymbol{z}, y \mid x; \theta)]$. Thus, when $w(\boldsymbol{z}) \propto r(y)$, we have $\theta \approx \hat{\theta}$. That is, under mild assumptions, the two optimization objectives are theoretically equivalent.

*In summary*, for a non-negative metric or reward $r(\cdot)$ that encourages the model to generate high-quality, correct answers, training with the vanilla objective also maximizes that the evaluation metric in downstream tasks.

### C.3.4 Experimental results

To validate the theoretical analysis presented above, we implement the goal-oriented objective to train the LLM. Following prior work, such as DeepSeek-R1 [17], we adopt two rule-based metrics, namely Exact Match (EM) and Accuracy (Acc.), as the function $r(y)$ in Proposition 1. These metrics are commonly used in open-domain QA and provide direct, interpretable supervision.

Figure 6 presents the performance of models trained using both the vanilla objective and the extended goal-oriented objective. We observe that models trained with different objectives (e.g., answer log-likelihood or specific metrics) converge to similar performances on the corresponding metric. For instance, the curves labeled *vanilla* and *w/ Exact Match* on the left side of Figure 6 both achieve a score of 50 in exact match after 5 iterations of training. A similar trend can also be found on the right side of Figure 6, where both the curves labeled *vanilla* and *w/ Accuracy* achieve a score of 53 in accuracy. This observation supports and aligns with the theoretical analysis presented above.

Additionally, we observe that the model trained using the vanilla answer log-likelihood converges faster and achieves the best average performance across both exact match and accuracy metrics. We analyze that there are two potential reasons. First, the $p(y \mid \boldsymbol{z}, x; \theta)$ is proportional to exact match and accuracy, which enhances the probability of generating correct answers, thus improving both metrics. Second, compared to rule-based metric that only evaluate the final result, $p(y \mid \boldsymbol{z}, x; \theta)$ provides a denser and continuous signal. This offers more effective guidance on how intermediate steps in the reasoning process influence the final outcome, allowing the model to fine-tune its reasoning trajectory toward the optimal solution.

### C.3.5 Detailed derivation for Proposition 1

Let $r(y) \geq 0$ be an evaluation metric, and define the expected objective as $\mathcal{J}(\theta) :=$ $\mathbb{E}_{(\boldsymbol{z},y) \sim p(\boldsymbol{z},y|x;\theta)}[r(y)]$. For any proposal distribution $q(\boldsymbol{z}, y)$ over reasoning-answer trajectories, we can construct a tractable evidence lower bound (ELBO) as:

$$\text{ELBO}(\theta, q) := \sum_{\boldsymbol{z},y} q(\boldsymbol{z}, y \mid x) \log \frac{r(y)p(\boldsymbol{z}, y \mid x; \theta)}{q(\boldsymbol{z}, y \mid x)}. \tag{32}$$

To prove Proposition 1, we show that there exists a constant $c > 0$ such that:

$$\left\| \arg\max_\theta \text{ELBO}(\theta, q) - \arg\max_\theta \mathcal{J}(\theta) \right\| \leq c \cdot (\text{KL}(q \parallel r \cdot p_\theta))^{1/2}, \tag{33}$$

where $r \cdot p_\theta$ denotes the unnormalized target distribution $q^*(\boldsymbol{z}, y) \propto r(y)p(\boldsymbol{z}, y \mid x; \theta)$.

*Proof.* We first apply Jensen's inequality to construct the ELBO:

$$\begin{aligned}
\mathcal{J}(\theta) = \sum_{\boldsymbol{z},y} r(y)p(\boldsymbol{z}, y \mid x; \theta) &= \sum_{\boldsymbol{z},y} q(\boldsymbol{z}, y \mid x) \cdot \frac{r(y)p(\boldsymbol{z}, y \mid x; \theta)}{q(\boldsymbol{z}, y \mid x)} \\
&\geq \exp\left( \sum_{\boldsymbol{z},y} q(\boldsymbol{z}, y \mid x) \log \frac{r(y)p(\boldsymbol{z}, y \mid x; \theta)}{q(\boldsymbol{z}, y \mid x)} \right) = \exp(\text{ELBO}(\theta, q)).
\end{aligned} \tag{34}$$

Taking logarithms (which preserves ordering due to its monotonicity), we obtain $\log \mathcal{J}(\theta) \geq \text{ELBO}(\theta, q)$ and note that $\arg\max_\theta \mathcal{J}(\theta) = \arg\max_\theta \log \mathcal{J}(\theta)$ due to the monotonicity of $\log(\cdot)$. To relate the maximization of $\text{ELBO}(\theta, q)$ and $\mathcal{J}(\theta)$, we begin by denoting $\theta^* := \arg\max_\theta \log \mathcal{J}(\theta) = \arg\max_\theta \mathcal{J}(\theta)$ and $\hat{\theta} := \arg\max_\theta \text{ELBO}(\theta, q)$. We aim to bound the distance $\|\theta^* - \hat{\theta}\|$.

We assume the objective function $\log \mathcal{J}(\theta)$ is $L$-smooth, i.e., it has $L$-Lipschitz continuous gradients[2]. This implies that for any $\theta_1, \theta_2$:

$$\log \mathcal{J}(\theta_1) \leq \log \mathcal{J}(\theta_2) + \nabla \log \mathcal{J}(\theta_2)^\top (\theta_1 - \theta_2) + \frac{L}{2} \|\theta_1 - \theta_2\|^2. \tag{35}$$

Applying this inequality with $\theta_1 = \theta^*$ and $\theta_2 = \hat{\theta}$, and using the fact that $\nabla \text{ELBO}(\hat{\theta}, q) = 0$ at the maximizer $\hat{\theta}$, we can write:

$$\begin{aligned}
\log \mathcal{J}(\theta^*) &\leq \log \mathcal{J}(\hat{\theta}) + \nabla \log \mathcal{J}(\hat{\theta})^\top (\theta^* - \hat{\theta}) + \frac{L}{2} \|\theta^* - \hat{\theta}\|^2 \\
&\leq \log \mathcal{J}(\hat{\theta}) + \frac{L}{2} \|\theta^* - \hat{\theta}\|^2,
\end{aligned} \tag{36}$$

where the last inequality assumes that the inner product term vanishes at first-order optimality. We now relate $\log \mathcal{J}(\hat{\theta})$ and $\text{ELBO}(\hat{\theta}, q)$ via:

$$\log \mathcal{J}(\hat{\theta}) - \text{ELBO}(\hat{\theta}, q) = \text{KL}(q \parallel r(y) \cdot p(\boldsymbol{z}, y \mid x; \hat{\theta})) + \log \mathcal{Z} \tag{37}$$

where $\mathcal{Z} = \sum_{\boldsymbol{z},y} r(y)p(\boldsymbol{z}, y \mid x; \hat{\theta}) = \mathcal{J}(\hat{\theta})$, so the gap equals:

$$\log \mathcal{J}(\hat{\theta}) - \text{ELBO}(\hat{\theta}, q) = \text{KL}(q \parallel q^*) \tag{38}$$

with $q^*(\boldsymbol{z}, y) \propto r(y)p(\boldsymbol{z}, y \mid x; \hat{\theta})$.

Putting it all together, we can obtain the following equation:

$$\log \mathcal{J}(\theta^*) - \text{ELBO}(\hat{\theta}, q) \leq \frac{L}{2} \|\theta^* - \hat{\theta}\|^2 + \text{KL}(q \parallel q^*) \tag{39}$$

By rearranging this, we have:

$$\|\theta^* - \hat{\theta}\|^2 \leq \frac{2}{L} \cdot \left( \log \mathcal{J}(\theta^*) - \text{ELBO}(\hat{\theta}, q) \right) \leq \frac{2}{L} \cdot \text{KL}(q \parallel r \cdot p_{\hat{\theta}}) \tag{40}$$

---

[2]https://en.wikipedia.org/wiki/Lipschitz_continuity

Taking square roots gives:

$$\|\theta^* - \hat{\theta}\| \le c \cdot \left(\mathrm{KL}(q \,\|\, r \cdot p_{\hat{\theta}})\right)^{1/2}, \tag{41}$$

where $c = \sqrt{\frac{2}{L}}$. This completes the proof. $\qquad\square$

*In summary*, this proposition suggests that we can maximize ELBO as a surrogate for $\mathcal{J}$. Below, we examine when this bound is tight and show that optimizing ELBO under a specific posterior $q^*$ is equivalent to maximizing the expected objective.

**Lemma C.4** (Tightness of the lower bound). *The lower bound in Eq. 34 becomes tight if and only if the proposal distribution satisfies:*

$$q(\boldsymbol{z}, y \mid x) = q^*(\boldsymbol{z}, y \mid x) \propto r(y) \cdot p(\boldsymbol{z}, y \mid x; \theta) \tag{42}$$

*Proof.* Equality in *Jensen's inequality* holds when the log argument is constant over $\boldsymbol{z}, y$:

$$\frac{r(y)p(\boldsymbol{z}, y \mid x; \theta)}{q(\boldsymbol{z}, y \mid x)} = \text{const.}, \quad \forall \boldsymbol{z}, y. \tag{43}$$

Solving for $q(\boldsymbol{z}, y \mid x)$ yields:

$$q^*(\boldsymbol{z}, y \mid x) = \frac{r(y)p(\boldsymbol{z}, y \mid x; \theta)}{\mathcal{Z}}, \quad \mathcal{Z} = \sum_{\boldsymbol{z}, y} r(y)p(\boldsymbol{z}, y \mid x; \theta) \tag{44}$$

i.e., the normalized metric-weighted joint distribution. $\qquad\square$

**Lemma C.5** (Optimality of ELBO maximization). *If $q = q^*(\boldsymbol{z}, y \mid x) \propto r(y)p(\boldsymbol{z}, y \mid x; \theta)$, then:*

$$\arg\max_{\theta} ELBO(\theta, q^*) = \arg\max_{\theta} \log \mathcal{J}(\theta) \tag{45}$$

*Proof.* From Lemma C.4, if $q = q^*$, then:

$$\mathrm{ELBO}(\theta, q^*) = \log \mathcal{J}(\theta) \implies \arg\max_{\theta} \mathrm{ELBO}(\theta, q^*) = \arg\max_{\theta} \mathcal{J}(\theta) \tag{46}$$

Thus, maximizing the ELBO is equivalent to maximizing the expected objective. $\qquad\square$

# D  More Experimental Details

## D.1  Experimental datasets

Following prior work [37, 38, 88, 90], we evaluate our method on a wide range of knowledge-intensive benchmarks, including Natural Questions (NQ) [36], HotpotQA [84], MuSiQue [70], and 2WikiMultihopQA (2WikiQA) [20]. Table 5 summarizes key statistics of these experimental datasets.

## D.2  Evaluation metrics

Following previous studies [57, 80, 88], we use the following metrics from KILT [49] for evaluation: *F1*, *Exact Match* (EM), and *Accuracy* (Acc.). *Exact Match (EM)* checks whether the predicted string exactly matches the ground truth. *Accuracy* checks whether the ground truth answer is included in the generated answer, often referred to as cover-Exact Match. *F1 Score* measures the overlap between the generated answer and the ground truth answer. It represents the harmonic mean of token-level precision and recall between these two sequences.

## D.3  Baselines

For a comprehensive evaluation, we compare EXSEARCH with a range of competitive baselines, categorized into three groups based on their use of retrieval and reasoning integration strategies: (i) direct reasoning without retrieval; (ii) advanced retrieval-augmented generation; and (iii) iterative retrieval-augmented generation.

Table 5: Statistics of our experimental datasets, where we provide the amount of training and evaluation dataset, the average length of input query (word) as well as the retrieval corpus.

| Experimental benchmarks | Training data size | Query length (Train) | Evaluation data size | Query length (Evaluation) | Retrieval corpus |
|---|---|---|---|---|---|
| Nature Question [36] | 58,622 | 9.21 | 6,489 | 9.16 | Wiki2018 |
| Hotpot QA [84] | 90,185 | 17.85 | 7,384 | 15.63 | Wiki2018 |
| MusiQue QA [70] | 19,938 | 15.96 | 2,417 | 18.11 | Wiki2018 |
| 2WikiMultiHopQA [20] | 167,454 | 12.74 | 12,576 | 11.97 | Wiki2018 |

**Direct reasoning without retrieval.** These methods rely solely on the LLM's internal parametric knowledge to reason over the input and generate answers, without incorporating any external information. We evaluate several recently released models, including both closed-source models, such as GPT-4o [48] and GPT-3.5 [47], as well as strong open-source models, such as DeepSeek-R1 [17], Qwen2.5 [83], QwQ-32B [68], LLaMA-3.3-70B [13], and Mistral-8x7B [26]. All of these models exhibit strong instruction-following and chain-of-thought reasoning capabilities, achieving remarkable performance on a wide range of natural language processing tasks.

**Advanced retrieval-augmented generation.** These models retrieve relevant documents from an external corpus, followed by optional document filtering mechanisms such as re-ranking or summarization, and concatenate the useful information into the LLM's input context for answer generation. Specifically, we evaluate several widely used approaches, including: (i) ChatQA [42] and RankRAG [88], which unify various knowledge-intensive tasks (e.g., knowledge-grounded dialogue, document re-ranking, question answering) into a single framework, training LLMs on large-scale datasets; (ii) InstructRAG [74], which inserts retrieved documents into the LLM's input and trains the model to generate chains of thought to identify useful content for answer generation; (iii) RetRobust [87], which trains the LLM to generate accurate answers conditioned on documents containing both correct and distracting information; (iv) Recomp [78], which uses extractive or abstractive summarization modules to filter out irrelevant content from retrieved documents, employing a large model (e.g., Flan-UL2, 20B)[3] to generate answers from the remaining information. In our experiments, we use the abstractive summarization module, as it serves as a more competitive baseline than its extractive counterpart.

**Iterative retrieval-augmented generation.** These approaches allow LLMs to actively interact with retrieval modules as needed. For a comprehensive evaluation, we include the following methods: (i) GenGround [64], which allows the LLM to iteratively retrieve external documents and refine its generated answer; (ii) DSPy [32], a programming framework that enables an LLM to decompose input queries and call external retrievers through structured prompting; (iii) SearchChain [80], which guides the LLM to generate a chain of queries and invoke retrieval at each step; (iv) Iter-RetGen [61] and IRCoT [71], which prompt the LLM to interact with the retriever in an iterative, few-shot manner; (v) Verify-and-Edit [94], which adaptively determines when to stop retrieval and finalize the answer based on the generation logits of the LLM; (vi) Generator-Retriever-Generator [1] (abbreviated as *Gen-Ret-Gen*), which instructs the LLM to answer a question using both model-generated and retrieved documents concatenated as context. All of the above methods are implemented on GPT-3.5, following their officially released reproducible settings. In addition, we include the following open-source baselines: (i) Search-o1 [39], which prompts the LLM to interleave query decomposition and document retrieval steps iteratively; (ii) Search-R1 [28], which trains the LLM to use external search engines via outcome rewards and Proximal Policy Optimization (PPO [60]); (iii) Self-RAG [4], which trains an LLM to retrieve documents on demand, assess their relevance, and generate final answers. This method is fine-tuned on 170k synthetic examples generated by a proprietary LLM.

### D.4 Data collection for warm-Up

**Data collection procedure.** To construct high-quality training trajectories that reflect realistic search behavior, we design a two-stage process: (1) first, we use GPT-4o to simulate step-by-step reasoning traces in the form of interleaved sub-queries; (2) then, each sub-query is paired with a

---

[3]https://huggingface.co/google/flan-ul2

retrieved document, simulating the retrieval process of a real system rather than directly relying on officially annotated golden documents.

To synthesize pseudo training trajectories, we leverage existing datasets, such as HotpotQA [84], to generate pseudo training data for our method, as it is a widely used multi-hop QA dataset similar to the setting of our agentic search method. In HotpotQA, each example consists of a complex multi-hop query, a final answer, and a set of officially annotated sub-queries along with their corresponding supporting documents. For each question, we prompt GPT-4o to simulate a step-by-step reasoning process that interleaves sub-query generation, document retrieval, and intermediate evidence extraction. Specifically, we instruct the model to act as an intelligent search agent. Given the original multi-hop question, its gold answer, and the full set of supporting Wikipedia passages, the model is asked to restructure the reasoning path into an interleaved sequence of three special operations:

- <THINK> to denote a generated sub-query;

- <SEARCH> to indicate the citation ID of the passage used to answer the sub-query; and

- <RECORD> to provide the answer to the sub-query.

The output continues in this format until the final answer is reached, which is prefixed by a <Final> tag. An example is included in the prompt to illustrate the expected format. This approach generates high-quality, interpretable reasoning trajectories that align with the structure of our method. All synthesized outputs are included in the supplementary material for reference.

```
You are an intelligent search agent that can simulate the question-
answering process based on my question and answer.

Given an open-domain query about Wikipedia, I have already marked the
correct answer at the end of the question and provided all the reference
Wikipedia passages needed to answer the question.
Your task is to reformat my provided question and references into a
detailed question-answering process.
Specifically, there should be three types of special tokens in your
output:
1. <THINK>, followed by a sub-query
2. <SEARCH>, followed by the citation ID
3. <RECORD>, followed by the answer to the sub-query

Since this is a multi-hop question, your output should interleave the `<
THINK>`, `<SEARCH>`, and `<RECORD>` tokens until you reach the final
answer.
Please start with a special token `<Final>` followed by the final answer.

Here is a concrete example to demonstrate the output format:
```example
Question: Which magazine was started first, Arthur's Magazine or First
for Women? (Answer: Arthur's Magazine)

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

```

```
<Final> Arthur's Magazine
```

Starting below, for the question "{question}", please complete your
output following the above requirements.

Question: {question}
Reference:
{golden doc}
Your Output:
```

After obtaining the simulated reasoning trajectories from GPT-4o, we pair each generated sub-query (`<THINK>` entry) with a retrieved document. To maintain consistency with our main experiments, we use the same retrieval setup, i.e., ColBERTv2.0 as the retriever and the 2018 Wikipedia dump as the retrieval corpus. For each sub-query, we first locate its corresponding gold supporting document (provided in the official HotpotQA dataset) and use it as a reference. We then apply ColBERTv2.0 to retrieve the most similar document from the full corpus based on its proximity to the gold reference. This ensures that the retrieved documents closely reflect what a real system would retrieve while remaining grounded in the original supervision signal. The resulting data consists of interleaved sub-queries and paired retrieved documents, effectively mimicking real multi-hop retrieval trajectories. All synthesized trajectories and retrieval pairs are included in the supplementary material.

**Cost for data collection.** The primary cost of our data synthesis arises from using GPT-4o to transform human-annotated sub-query paths into simulated search trajectories, following the pattern used in EXSEARCH. In our main experiment, we generate 1,000 examples, with an average input length of 5,095.29 tokens and an average output length of 732.20 tokens. Based on OpenAI's GPT-4o pricing[4], the total cost for constructing these 1,000 examples is approximately \$12.73 for input and \$7.30 for output, totaling around \$20. *Thus, the cost per example is* $\$\frac{20}{1000} = 0.02$. Token counts are directly obtained from OpenAI's API call messages. See the official OpenAI API documentation[5] for more details.

### D.5   Implementation details and hyperparameters

During the training stage, we trained the models with a learning rate of $2 \times 10^{-6}$, using DeepSpeed Zero 3 for efficient distributed optimization. The batch size was set to 4 for the 3B, 7B, and 8B models, and reduced to 2 for the 24B model due to memory constraints. We applied a linear warm-up (10% of total steps), followed by a cosine learning rate scheduler. All experiments used BF16 mixed-precision training with a sequence length cutoff of 8192 tokens. For each training example, we sampled 2 search trajectories (We also experimented with varying the sampling number, choosing $\{1, 2, 4, 6, 8\}$, but observed no significant difference in final performance). Table 6 summarizes the hyperparameters in our implementation. The models used in our experiments can be directly downloaded from HuggingFace, an open-source platform for machine learning and deep learning.

During the inference stage, the maximal step $T$ for the *thinking* → *search* → *recording* iteration is set to 5.

Table 6: Experimental settings for model training.

| Model | Batch size | Learning rate | Cutoff length | Scheduler | Gradient accumulation |
|-------|-----------|---------------|---------------|-----------|-----------------------|
| Qwen-2.5-3B-instruct | 4 | $2 \times 10^{-6}$ | 8192 tokens | Cosine | 16 |
| Qwen-2.5-7B-instruct | 4 | $2 \times 10^{-6}$ | 8192 tokens | Cosine | 16 |
| Llama-3.2-3B-instruct | 4 | $2 \times 10^{-6}$ | 8192 tokens | Cosine | 16 |
| Llama-3.1-8B-instruct | 4 | $2 \times 10^{-6}$ | 8192 tokens | Cosine | 16 |
| Mistral-7B-instruct-v0.3 | 4 | $2 \times 10^{-6}$ | 8192 tokens | Cosine | 16 |
| Mistral-24B-small-2501 | 2 | $2 \times 10^{-6}$ | 8192 tokens | Cosine | 16 |

---

[4]https://platform.openai.com/docs/pricing
[5]https://platform.openai.com/docs/api-reference/introduction

The training process of EXSEARCH is relatively straightforward to implement. Skeleton PyTorch code for EXSEARCH is demonstrated below.

```python
import torch
import torch.nn.functional as F

def compute_causal_weight(ref_model, tokenizer, trajectory, answer):
    """
    Compute the reward weight w(z) = p(answer | trajectory).

    This function estimates how likely the reference model (at step t) would
    produce the correct answer y given a sampled reasoning trajectory z.

    Args:
        ref_model: The frozen language model at current iteration.
        tokenizer: The tokenizer corresponding to the model.
        trajectory: A list of messages (e.g., in chat format).
        answer: The gold answer string.

    Returns:
        weight (float): Estimated likelihood p(y | x, z), as a scalar reward.
    """
    input_ids = tokenizer.apply_chat_template(trajectory,
                                              return_tensors="pt").input_ids
    label_ids = tokenizer(answer, return_tensors="pt").input_ids

    with torch.no_grad():
        logits = ref_model(input_ids).logits[:, -label_ids.size(1):, :]
        log_probs = F.log_softmax(logits, dim=-1)
        answer_logp = torch.gather(log_probs, 2, label_ids.unsqueeze(-1))
        answer_logp = answer_logp.squeeze(-1)
        weight = answer_logp.sum().exp().item()  # Likelihood as scalar
    return weight

def M_step_learning(ref_model, input_ids, label_ids, weights):
    """
    Compute re-weighted cross-entropy loss for a given trajectory.

    This function performs the forward computation of the M-step by applying
    a scalar reward to the log-likelihood loss, encouraging trajectories
    that lead to correct answers.

    Args:
        ref_model: The language model to be updated.
        input_ids: The sampled trajectories (token IDs), shape (B, L)
        label_ids: The gold answers (token IDs), shape (B, L)
        weights: Pre-computed weights, shape (B)

    Returns:
        loss (Tensor): A scalar loss value used for gradient update.
    """
    logits = ref_model(input_ids).logits[:, :-1, :]
    labels = label_ids[:, 1:].to(logits.device)

    loss_fct = torch.nn.CrossEntropyLoss(ignore_index=-100, reduction="mean")
    loss = loss_fct(logits.view(-1, logits.size(-1)), labels.view(-1))

    # Apply pre-computed weights
    weights = torch.exp(weights - weights.max()) / weights.sum()
```

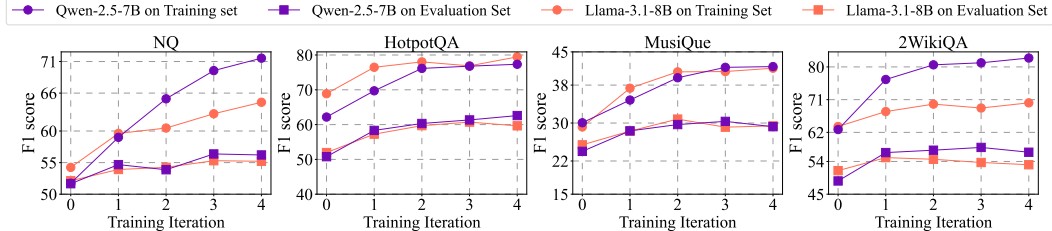

Figure 7: Training convergence of Qwen-2.5-7B and Llama-3.1-8B, where we report the *F1* score for checkpoints in each iteration. The 0th iteration indicates the initial warm-up training.

```
loss = loss * weights.mean()  # Weight the loss by the average of the weights
return loss
```

Table 7: Precision@$K$ ($K = 3, 5$) for our method (*w/* 8B Llama3.1 and 7B Qwen2.5) and strong baselines.

| Tasks | NQ | | HotpotQA | | MusiQue | | 2WikiQA | | Avg. | |
|---|---|---|---|---|---|---|---|---|---|---|
| **Metrics** | P@3 | P@5 | P@3 | P@5 | P@3 | P@5 | P@3 | P@5 | P@3 | P@5 |
| ColBERTv2.0 | 41.36 | 35.97 | 28.71 | 25.20 | 6.41 | 5.44 | 17.96 | 16.10 | 23.61 | 20.68 |
| *Re-ranking* | | | | | | | | | | |
| MonoT5 [46] | 42.44 | 36.73 | 31.48 | 26.86 | 6.62 | 5.85 | 19.27 | 16.93 | 24.95 | 21.59 |
| BGE [76] | 49.55 | 40.29 | 34.50 | 28.67 | 7.70 | 6.50 | 20.45 | 18.00 | 28.05 | 23.36 |
| RankVicuna [51] | 42.68 | 36.97 | 29.49 | 26.71 | 7.09 | 6.02 | 18.75 | 17.71 | 24.50 | 21.85 |
| RankZepyhr [51] | 41.54 | 37.77 | 30.76 | 27.35 | 8.98 | 7.82 | 18.87 | 16.57 | 25.04 | 22.38 |
| *Query decomposition* | | | | | | | | | | |
| Search-o1 [39] | 42.42 | 37.01 | 42.01 | 34.11 | 17.23 | 13.13 | 25.34 | 18.24 | 31.75 | 25.62 |
| Search-r1 [28] | 40.17 | 36.86 | 27.12 | 32.67 | 4.07 | 8.30 | 16.25 | 23.49 | 21.90 | 25.33 |
| Ours-Qwen2.5-7B | 43.50 | 37.58 | 44.47 | 37.49 | 18.85 | 15.91 | 29.60 | 23.33 | 34.11 | 28.58 |
| Ours-Llama3.1-8B | 43.56 | 37.49 | 43.76 | 36.91 | 19.33 | 16.30 | 28.22 | 22.20 | 33.72 | 28.22 |

## D.6 Supplementary experimental results

**Supplementary results for retrieval performance.** In the main body of our paper, we have report the recall@K score for the proposed EXSEARCH and strong baselines. Below, we supplements the performance in terms of precision@K score in Table 7. These results further indicate that EXSEARCH, by dynamically expanding the search as reasoning unfolds, can also improve retrieval performance in addition to enhancing end-to-end answer accuracy.

**Supplementary results for training convergence experiments.** The theoretical analysis in § C.2 guarantees that our training objective is non-decreasing across iterations. To empirically validate this, we evaluate model checkpoints after each iteration on both the training and test sets. In the main body of our paper, we report the performance in terms of the exact match scores. In this appendix, we further supplements the performance in terms of F1 metrics in Figure 7 for a more comprehensive comparison. We observe consistent performance improvement over iterations, eventually stabilizing, confirming the expected non-decreasing convergence. On evaluation sets, models typically peak by the second or third iteration, demonstrating rapid practical convergence. These results align with our theoretical guarantees and highlight the efficiency of our approach.

**Supplementary results for EXSEARCH with cold start.** To investigate the role of warm-up supervision, we train separate models with varying amounts of supervised fine-tuning (SFT) data, denoted by $K$, and apply EXSEARCH to each independently. The exact match (EM) scores has been reported in the main body of our paper as the primary evaluation metric. In this appendix, the corresponding F1 scores are presented in Figure 8 as a supplementary result. We observe that all models benefit from iterative self-training, with larger $K$ values leading to faster convergence and better final performance. Remarkably, even with only $K = 100$ or no warm-up supervision ($K = 0$), the models still achieve substantial gains, demonstrating the robustness of EXSEARCH in

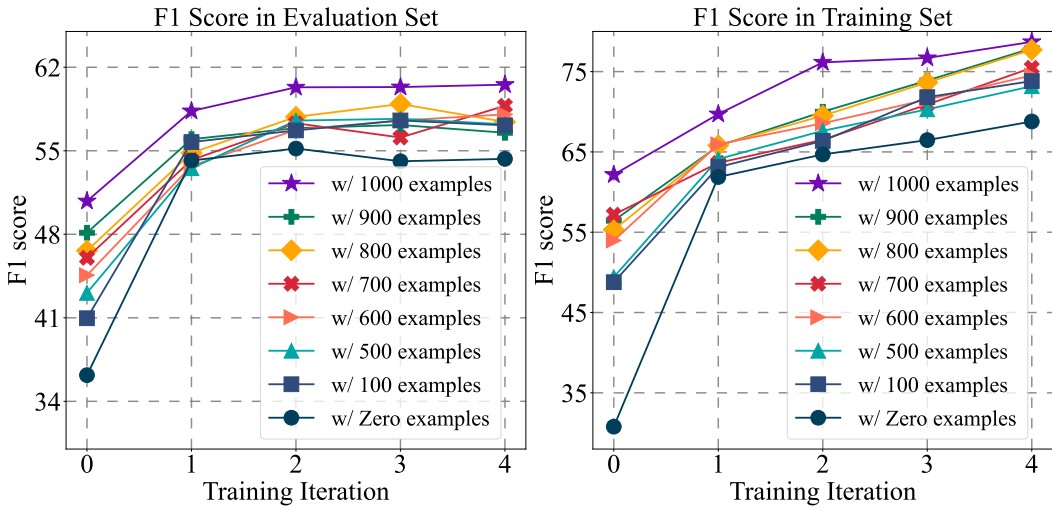

Figure 8: F1 score for ExSearch-Qwen-2.5-7B that was initially empowered by different amounts of warm-up data.

low-resource and cold-start scenarios. Based on the cost analysis in § D.4, synthesizing each example costs only $0.02 on average. Thus, empirically, in our experiment, synthesizing 100 examples incurs an approximate cost of $2.

**Supplementary results for human evaluation.** Considering the potential bias of automatic metrics [63], we conduct a human evaluation with three educated individuals assessing the *correctness* of 100 randomly sampled cases from five benchmarks, using a two-scale rating (1 for correct; 0 for incorrect). Each query is paired with the corresponding golden documents and ground truth answers from the original datasets, which serve as references for the human evaluators. We ask at least two annotators to evaluate the same case repeatedly. If there is a discrepancy between two annotators, ask a third annotator to recheck it. The results are presented in Table 8, where we found that our method achieve the highest correctness, further indicating its effectiveness. In our human evaluation, the overall Kappa value is 0.771, demonstrating substantial agreement among the annotators. This indicates the reliability of the evaluation process.

Table 8: Human evaluation on 100 randomly sampled cases.

|  | **GPT-4o** | **InstructRAG** | **Search-o1** | **Search-R1** | **ExSearch** |
|---|---|---|---|---|---|
| **Correctness** | 48/100 | 40/100 | 46/100 | 50/100 | 54/100 |

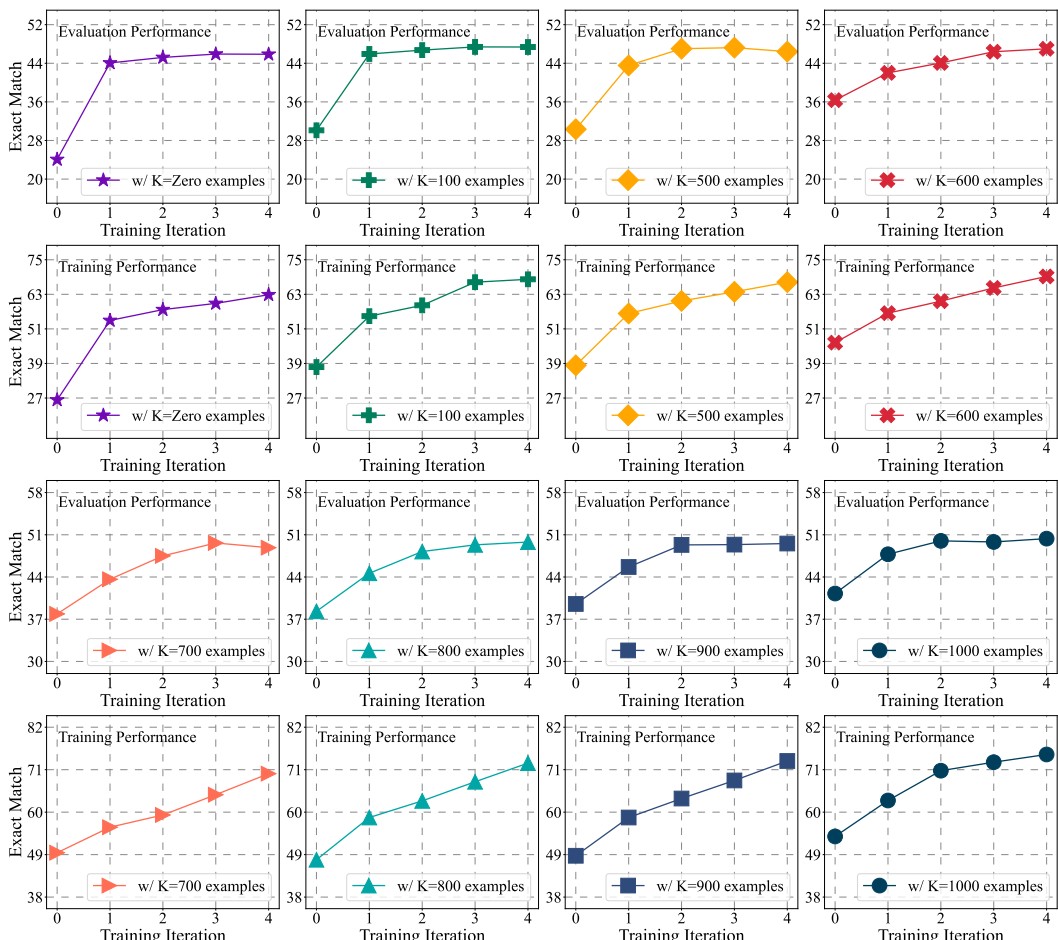

Figure 9: Exact match score for EXSEARCH-Qwen-2.5-7B, which is initially empowered by varying amounts of warm-up data, during the iterative training process in EXSEARCH.

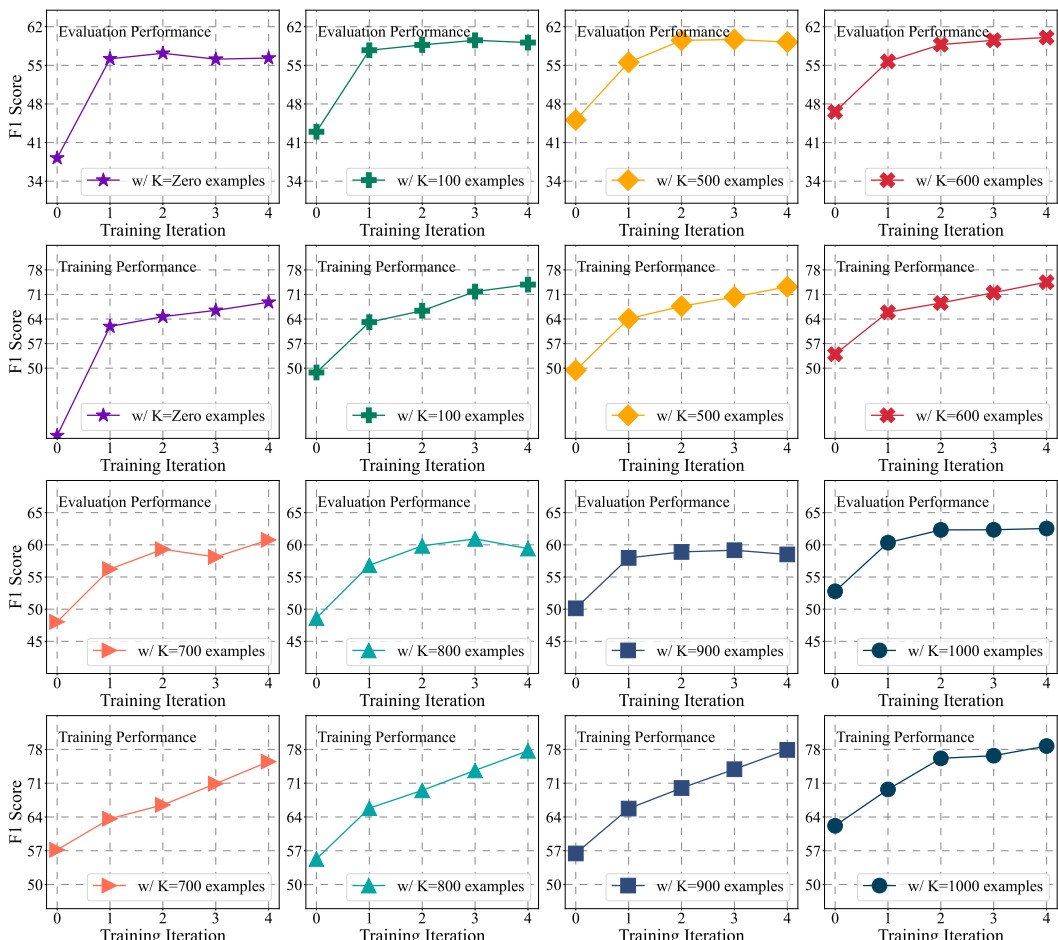

Figure 10: F1 score for EXSEARCH-Qwen-2.5-7B, which is initially empowered by varying amounts of warm-up data, during the iterative training process in EXSEARCH.

Table 9: Experimental results for applying our method to various LLMs.

| Tasks | NQ | | | HotpotQA | | | MuSiQue | | | 2WikiQA | | | Avg. | | |
|---|---|---|---|---|---|---|---|---|---|---|---|---|---|---|---|
| Metrics | F1 | EM | Acc. | F1 | EM | Acc. | F1 | EM | Acc. | F1 | EM | Acc. | F1 | EM | Acc. |
| Ours-Qwen-2.5-3B | 46.23 | 36.76 | 39.12 | 54.32 | 42.22 | 46.08 | 19.44 | 13.76 | 13.94 | 43.39 | 37.24 | 44.78 | 40.85 | 32.50 | 35.98 |
| Ours-Qwen-2.5-7B | 56.37 | 47.07 | 51.75 | 62.59 | 50.35 | 54.32 | 29.68 | 22.03 | 24.34 | 57.14 | 52.62 | 54.37 | 51.45 | 43.02 | 46.20 |
| Ours-Llama-3.3-3B | 41.42 | 33.49 | 35.17 | 44.12 | 33.53 | 36.14 | 17.64 | 11.23 | 11.82 | 41.73 | 36.28 | 42.22 | 36.23 | 28.63 | 31.34 |
| Ours-Llama-3.1-8B | 55.21 | 43.71 | 50.76 | 60.72 | 47.59 | 53.59 | 30.83 | 20.98 | 24.65 | 54.62 | 47.48 | 54.21 | 50.35 | 39.94 | 45.80 |
| Ours-Mistral-7B-instruct | 56.83 | 45.13 | 52.05 | 59.65 | 50.35 | 54.78 | 30.32 | 23.47 | 24.98 | 53.93 | 47.38 | 54.82 | 50.18 | 41.58 | 46.66 |
| Ours-Mistral-2501-24B | 59.89 | 47.62 | 56.39 | 67.03 | 54.51 | 59.98 | 35.84 | 23.68 | 28.54 | 60.81 | 53.19 | 61.59 | 55.89 | 44.75 | 51.63 |

# E   EXSEARCH-Zoo: Extending EXSEARCH for Diverse Scenarios

Extensive experiments on a wide range of knowledge-intensive benchmarks demonstrate the state-of-the-art performance of EXSEARCH, as detailed in the main body of the paper. This strong performance motivates us to extend EXSEARCH to more diverse scenarios. We introduce EXSEARCH-Zoo, a comprehensive framework that enhances EXSEARCH along two key dimensions: (i) Diverse backbone LLMs across model families (LLaMA, Qwen, Mistral) and scales (7B-24B parameters); (ii) Extended actions, such as document re-ranking, to enrich the existing reasoning actions (*thinking*, *search*, and *recording*). Below, we describe how EXSEARCH is extended along each of these dimensions.

## E.1   Diverse model families and scales

We apply EXSEARCH to a range of LLMs with varying parameter sizes and model families. As shown in Table 9, the results exhibit a clear scaling-law pattern. More specifically, we highlight two key observations: First, there is a consistent performance improvement as the model size increases; Second, even models with as few as 3B parameters achieve strong performance when augmented with our method. These findings suggest that EXSEARCH is broadly applicable and scales favorably across different model families.

## E.2   Extended retrieval strategy

In EXSEARCH, when faced with complex information needs, the LLM iteratively infers missing knowledge, evaluates retrieved evidence, and adapts its search strategies as new information is acquired. This behavior is formalized as a reasoning-interleaved search trajectory consisting of three core actions: **think → seek → record**, as introduced in § C.1. Although this pattern is general, for more complex tasks, we may also need to introduce additional actions to improve the end-to-end performance. To illustrate the extensibility of EXSEARCH, we introduce an additional document re-ranking action to the vanilla EXSEARCH framework. The re-ranking step acts as a filtering mechanism, allowing the model to discard irrelevant content and focus on cleaner, more useful evidence before generating intermediate reasoning steps.

**Example: Re-ranking as a reasoning action.**   We introduce a document re-ranking step between retrieval and evidence selection, resulting in a four-step reasoning pattern: **think → seek → rank → record**. Specifically, *we implement this re-ranking following generative re-ranking techniques [51, 67], where the LLM reads the retrieved documents and autoregressively generates a ranked list of selected identifiers (e.g.,* [1] > [2] > [3]*)*.

The updated reasoning process consists of: (1) generating a sub-query $x_i$; (2) retrieving candidate documents $\boldsymbol{d}_i$ using the retriever $\mathcal{R}$; (3) re-ranking the retrieved documents to select the most relevant ones, denoted as $\hat{\boldsymbol{d}}_i$; and (4) reflecting on the selected documents $\hat{\boldsymbol{d}}_i$ by extracting an intermediate answer $e_i$ to the sub-query $x_i$.

Formally, we represent the full trajectory as a sequence of triplets $\boldsymbol{z} = \{(x_i, \hat{\boldsymbol{d}}_i, e_i)\}_{i=1}^{|\boldsymbol{z}|}$, and define its likelihood conditioned on input $x$ and parameters $\theta$ as:

$$
\begin{aligned}
p(\boldsymbol{z} \mid x; \theta) &= \prod_{i=1}^{|\boldsymbol{z}|} p((x_i, \hat{\boldsymbol{d}}_i, e_i) \mid x, \boldsymbol{z}_{<i}; \theta) \\
&= \prod_{i=1}^{|\boldsymbol{z}|} \underbrace{p(x_i \mid \boldsymbol{z}_{<i}; \theta)}_{\text{thinking}} \cdot \underbrace{p(\hat{\boldsymbol{d}}_i \mid x, \mathcal{R}(x_i); \theta)}_{\text{retrieval and re-ranking}} \cdot \underbrace{p(e_i \mid x_i, \hat{\boldsymbol{d}}_i; \theta)}_{\text{recording}}.
\end{aligned}
\tag{47}
$$

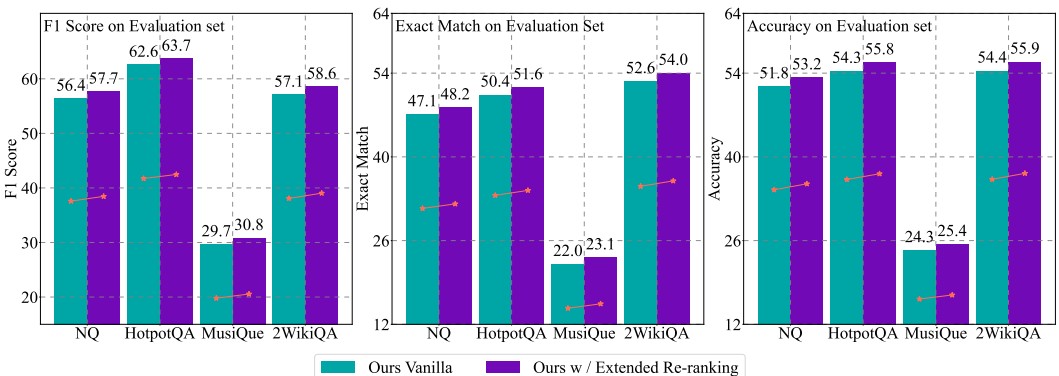

Figure 11: Performance of EXSEARCH when extended with an additional document re-ranking action, where we report the exact match score, F1 score and accuracy for a comprehensive comparison.

After finalizing a full trajectory $z$, the LLM generates a final answer $y$ based on all intermediate reasoning steps via $y \sim p(y \mid x, z; \theta)$.

Compared to the vanilla EXSEARCH, the only modification in the above variant is the additional document re-ranking action to the E-step, while the overall EM training framework remains unchanged. Thus, we can apply the same EM-style optimization in Eq. 19 for this variant, with the learning objective formulated as:

$$\theta = \arg\max_{\theta} \mathbb{E}_{z \sim p(z \mid x; \theta^t)} \left[ w(z) \log p(z \mid x; \theta) + w(z) \log p(y \mid x, z; \theta) \right], \quad (48)$$

where $w(z)$ is the importance weight based on how well $z$ supports the correct answer. The log-likelihood decomposes into two components: $\mathcal{L}_{\mathcal{R}} = \sum_{i=1}^{|z|} \log p(x_i \mid z_{<i}; \theta) + \log p(\hat{d}_i \mid x, \mathcal{R}(x_i); \theta) + \log p(e_i \mid x_i, \hat{d}_i; \theta)$ and $\mathcal{L}_{\mathcal{A}} = \log p(y \mid x, z; \theta)$. The former $\mathcal{L}_{\mathcal{R}}$ corresponds to learning iterative search, while the latter $\mathcal{L}_{\mathcal{A}}$ corresponds to learning how to aggregate information for answer generation.

**Experimental results.** Figure 11 reports the results of incorporating re-ranking into the reasoning trajectory of EXSEARCH. We observe that adding a re-ranking step consistently improves performance across all evaluated datasets. For example, on MuSiQue and 2WikiQA, we observe average gains of +1.9 and +2.4 points in F1 score, respectively, indicating that selective document filtering can further enhance the model's ability to reason over relevant information within EXSEARCH. These results highlight the extensibility of our framework: by augmenting the action space with an additional step, we can effectively adapt the search-reasoning loop to more complex settings. This enables us to move beyond fixed action templates and incorporate new reasoning operations based on the specific requirements of downstream tasks. We believe this opens avenues for extending EXSEARCH to specialized tasks, such as retrieval-augmented fact verification, open-domain multi-hop reasoning, or tool-integrated planning, by incorporating additional task-specific actions.

# F   Prompt and Case Study

## F.1   System prompt in EXSEARCH

To enable step-wise reasoning and evidence-aware retrieval, we design a structured system prompt for EXSEARCH that guides the model to simulate an intelligent search agent. The detailed content is shown below. This prompt instructs the model to decompose a complex query into sub-queries (*thinking*), retrieve relevant documents from Wikipedia based on each sub-query (*search*), and extract factual answers from the retrieved content (*recording*). This system prompt aims to encourage the LLM to perform multi-hop reasoning via an interleaved search-and-read loop, rather than attempting to answer the question in a single step. It also enforces an interpretable action trace, allowing us to diagnose the model's behavior at each stage of the reasoning process. Finally, the generation ends with a `<FINAL>` token and the final answer, enabling seamless integration into downstream pipelines and evaluation.

```
You are an intelligent search agent capable of simulating a question-
answering process by actively seeking information from Wikipedia to
answer a given question.

Specifically, given an open-domain query, please iteratively: (1)
Formulate a sub-query to search on Wikipedia; (2) Select useful documents
 from the search results and (3) Extract supporting facts from the
selected documents.
Your output should include three types of special actions corresponding
to the above steps:
(1) <THINK>: Formulate a sub-query.
(2) <SEARCH>: Retrieve and carefully read the documents using the
formulated sub-query.
(3) <RECORD>: Extract the answer to the sub-query from the documents.

Since this is a multi-hop question, your output should interleave <THINK
>, <SEARCH> and <RECORD> actions until reaching the final answer.
Conclude your output with the special token <FINIAL> followed by the
final answer.

Below is the task for you to complete:

<USER QUERY> {THE_INPUT_TASK}
Your Output:
```

## F.2 Human evaluation

In order to assess the end-to-end performance of the proposed model, we conducted human evaluations to verify whether the model's outputs align with the ground-truth answers. The evaluation procedure involves presenting human evaluators with a series of tasks, where they are asked to judge whether the model's prediction is consistent with the correct answer. The task provided to the human evaluators is as follows, where placeholders {question}, {model output}, and {answer} are replaced with the actual question, model's response, and the ground-truth answer, respectively. The evaluators are asked to determine whether the model's prediction implies the correct ground-truth answer.

> **Guidance Human to Evaluate the Correctness of a Model Output**
>
> In the following task, you are given a question, a model Prediction for the question, and a ground-truth Answer to the question. You should decide whether the model's Prediction implies the Ground-truth Answer.
>
> Question
> {question}
>
> Prediction
> {model output}
>
> Ground-truth Answer
> {answer}
>
> Does the Prediction imply the Ground-truth Answer? Output Yes or No:

To guarantee annotation quality, we ask at least two annotators to evaluate the same questions repeatedly. If there is a discrepancy between the two annotators (i.e., when two annotators give a different correctness score), a third annotator is asked to review and resolve the inconsistency. To quantify the reliability of the annotations, we compute Cohen's Kappa score to measure inter-annotator agreement. A Kappa value close to 1 indicates strong agreement, while a value near 0 suggests random agreement. In our human evaluation, the overall Kappa value is 0.771, demonstrating substantial agreement among the annotators. This indicates that our evaluation process is reliable and that the results can be confidently used to assess the model's performance.

### F.3 Case studies

**Good case: Correct runtime case of EXSEARCH.** This example demonstrates a successful reasoning trajectory executed by EXSEARCH for a compositional multi-hop question: *Navarone Garibaldi is the half-brother of a singer who has been married how many times?* The task requires (i) identifying the half-sibling relationship; and (ii) retrieving the number of marriages of the referenced singer. The model first identifies Lisa Marie Presley as Navarone Garibaldi's half-sister, then retrieves the number of times she has been married. It correctly terminates the search after the evidence is obtained and outputs the gold answer, i.e., four.

Below, in each block, the content behind `<THINK>` and `<RECORD>` represents model-generated outputs, while `<SEARCH>` denotes the retrieved document snippets from the external corpus.

```
INPUT QUERY: Navarone Garibaldi is the half-brother of a singer who has
been married how many times? (Golden answer: four times)
```

```
============ Turn 1 ============
<THINK> Who is Navarone Garibaldi's half-brother?
<SEARCH> [0] Title: Navarone Garibaldi. Content: Navarone Garibaldi
Navarone Anthony Garibaldi (born March 1, 1987) is an American musician.
He is the frontman of the band Them Guns. Garibaldi is the son of Marco
Garibaldi and Priscilla Presley, and also half-brother of Lisa Marie
Presley. Navarone Anthony Garibaldi was born to actress Priscilla Presley
 and Marco Garibaldi on March 1, 1987 in Santa Monica, California. With
his mother Priscilla being the former wife of the late Elvis Presley,
media scrutiny was often focused on the family. Garibaldi's half-sister
is Lisa Marie Presley, Elvis's only child. He is the uncle of Riley and
Benjamin Keough and Harper... [2]... [3]...
<RECORD> Lisa Marie Presley
```

```
============ Turn 2 ============
<THINK> How many times has Lisa Marie Presley been married?
<SEARCH> [0] Title: Lisa Marie Presley. Content: Lisa Marie Presley Lisa
Marie Presley (born February 1, 1968) is an American singer-songwriter.
She is the only child of singer and actor Elvis Presley and actress
Priscilla Presley, as well as the sole heir to her father's estate.
Presley has developed a career in the music business and has issued three
 albums. She has been married four times, including to singer Michael
Jackson and actor Nicolas Cage, before marrying music producer Michael
Lockwood, father of her twin girls. Lisa Marie was born on February 1,
1968, to Elvis and Priscilla Presley at Baptist Memorial Hospital in
Memphis, Tennessee, nine
<RECORD> four
```

```
============ Turn 3 (answer aggregation) ============
<FINAL>
<SEARCH> end search.
The final answer to the input question "Navarone Garibaldi is the half-
brother of a singer who has been married how many times?"
Output: four
```

**Bad case: Over-searching due to missed reasoning.** We present a concrete example below, where the LLM gives a correct answer but overly relies on the retrieval module. In Turn 1 and Turn 2, the model successfully retrieves the occupations of both individuals via two well-formed sub-queries. However, instead of reasoning over the retrieved facts to answer the question, the model repeatedly issues similar external queries (Turn 3–5), such as *Are there any shared occupations between Maher Sabry and Til Schweiger?* These repeated queries yield no new information and reflect an over-dependence on retrieval, rather than using previously acquired knowledge for local inference. This highlights a limitation in current retrieval-augmented models: they often fail to consolidate retrieved evidence before issuing additional queries, resulting in unnecessary search steps.

```
USER QUERY: What occupations are shared by Maher Sabry and Til Schweiger?
 (Golden answer: director, and producer)
```

```
============ Turn 1 ============
<THINK> What is Maher Sabry's occupation?
<SEARCH> [1] Title: Maher Sabry. Content: Maher Sabry is an Egyptian
theater director, playwright, film director, producer and screenwriter,
poet, writer and cartoonist. A gay activist...; [2]...; [3]...
<RECORD> Maher Sabry's occupations include theater director, playwright,
film director, producer, screenwriter, poet, writer, and cartoonist.
```

```
============ Turn 2 ============
<THINK> What is Til Schweiger's occupation?
<SEARCH> [1] Title: Til Schweiger. Content: Til Schweiger Tilman Valentin
 Til Schweiger (born 19 December 1963) is a German actor, voice actor,
film director, film producer, and film editor. He runs his own production
 company, ...; [2] ... [3] ...
<RECORD> Til Schweiger's occupations include actor, voice actor, film
director, film producer, and film editor.
```

```
============ Turn 3 ============
<THINK> Are there any shared occupations between Maher Sabry and Til
Schweiger?
<SEARCH> [1] Title: Maher Sabry. Content: Maher Sabry is an Egyptian
theater director, playwright, film director, producer and screenwriter,
poet, writer and cartoonist. A gay activist,... [2] ... [3] ...
<RECORD> Maher Sabry's occupations include theater director, playwright,
film director, producer, screenwriter, poet, writer, and cartoonist.
```

```
============ Turn 4 ============
<THINK> Are there any shared occupations between Maher Sabry and Til
Schweiger?
<SEARCH> [1] Title: Maher Sabry. Content: Maher Sabry is an Egyptian
theater director, playwright, film director, producer and screenwriter,
poet, writer and cartoonist. A gay activist,... [2] ... [3] ...
<RECORD> Maher Sabry's occupations include theater director, playwright,
film director, producer, screenwriter, poet, writer, and cartoonist.
```

```
============ Turn 5 ============
<THINK> Are there any shared occupations between Maher Sabry and Til
Schweiger?
<SEARCH> [1] Title: Maher Sabry. Content: Maher Sabry is an Egyptian
theater director, playwright, film director, producer and screenwriter,
poet, writer and cartoonist. A gay activist,... [2]... [3]...
<RECORD> Maher Sabry's occupations include theater director, playwright,
film director, producer, screenwriter, poet, writer, and cartoonist.
```

```
============ Turn 6 ============
<THINK> Are there any shared occupations between Maher Sabry and Til
Schweiger?
<SEARCH> [1] Title: Maher Sabry. Content: Maher Sabry is an Egyptian
theater director, playwright, film director, producer and screenwriter,
poet, writer and cartoonist. A gay activist,... [2] ... [3] ...
<RECORD> Maher Sabry's occupations include theater director, playwright,
film director, producer, screenwriter, poet, writer, and cartoonist.
```

```
============ Turn 7 ============
<THINK> Are there any shared occupations between Maher Sabry and Til
Schweiger?
```

```
<SEARCH> [1] Title: Maher Sabry. Content: Maher Sabry is an Egyptian
theater director, playwright, film director, producer and screenwriter,
poet, writer and cartoonist. A gay activist,... [2] ... [3] ...
<RECORD> Maher Sabry's occupations include theater director, playwright,
film director, producer, screenwriter, poet, writer, and cartoonist.
```

```
============ Turn 8 (Up to the maximum search times) ============
<FINAL> theater director, playwright, film director, producer,
screenwriter
please give the final answer to the input question: What occupations are
shared by Maher Sabry and Til Schweiger?
Playwright, film director, producer, screenwriter
```

**Bad case: Under-searching due to premature entity binding.**   In this example, the user asks
a seemingly straightforward compositional question: *Into Dust is a song by the alternative rock
band formed in which city?*  The correct answer is Santa Monica, California, as the song *Into
Dust* is performed by the alternative rock band Mazzy Star.  However, the model prematurely
assumes an entity match based on a partial overlap with the query phrase "Into Dust," mistakenly
linking it to a different band named *Cities in Dust*, a Canadian punk group. As shown in the first
retrieval result, this band has no association with the queried song but is erroneously selected due to
superficial string-level similarity and genre proximity. The model then terminates the search after
this single retrieval step and outputs Hamilton, Ontario, as the final answer, based solely on the
retrieved band's origin. This reflects a classic under-searching failure mode, where the model fails to
verify whether the retrieved entity is semantically compatible with the query. In this case, a simple
clarification query, such as "Who performed *Into Dust* (song)?" would have revealed the correct
band and corresponding city. This example highlights the need for disambiguation-aware reasoning
mechanisms that delay answer generation until entity identity is sufficiently grounded.

```
USER QUERY: Into Dust is a song by the alternative rock band formed in
which city? (Golden answer: Santa Monica, California)
```

```
==========
<THINK> Which city was the alternative rock band that performed "Into
Dust" formed in?
<SEARCH> [1] Title: Cities in Dust (band). Content: Week in Toronto.
Independent record label Paper Bag Records signed Cities in Dust in April
 2006, in part due to the onstage antics of Frank. Soon after, their
debut album Night Creatures was released. Cities in Dust appeared at the
World Electronic Music Festival in June 2006. The band was nominated for
three awards at the 2006 Hamilton Music Awards: Record of the Year, New
Artist/Group of the Year, Alternative Recording of the Year. Night
Creatures was named by Exclaim! magazine as one of the Top 10 Punk albums
 of 2006 in Canada. and it received heavy rotation on CBC
[2] ... [3]...
<RECORD> Hamilton, Ontario
```

```
USER QUERY: Into Dust is a song by the alternative rock band formed in
which city?
please give the final answer to the input question: Into Dust is a song
by the alternative rock band formed in which city?
Hamilton, Ontario
```

Table 10: Case study of under-searching caused by entity confusion and early termination.

| Aspect | Observation |
|---|---|
| **User Query** | Into Dust is a song by the alternative rock band formed in which city? |
| **Gold Answer** 
 **Retrieved Fact** | Santa Monica, California (Mazzy Star) 
 Hamilton, Ontario (based on Cities in Dust band) |
| **Search Count** | 1 |
| **Entity Linking Error** | Mistook "Into Dust" as a song by "Cities in Dust" |
| **Failure Mode** | Under-searching: Incorrect assumption based on first retrieved entity, no disambiguation step. |
| **Suggested Fix** | Add entity verification query: *"Who performed Into Dust?"* or *"Into Dust band name"* before inferring location. |

Table 11: Case study illustrating over-searching due to lack of intermediate reasoning.

| Aspect | Observation |
|---|---|
| **User Query** | What occupations are shared by Maher Sabry and Til Schweiger? |
| **Query Type** | Compositional: Requires comparing two entities' attributes. |
| **Sub-Queries Issued** | Turn 1: What is Maher Sabry's occupation? 
 Turn 2: What is Til Schweiger's occupation? |
| **Knowledge Retrieved** | All relevant occupations were retrieved correctly for both individuals by Turn 2. |
| **Expected Behavior** 
 **Observed Behavior** | The model should compare two sets of occupations and output the shared ones. 
 Model redundantly issues semantically similar queries in Turn 3–5, such as: 
 *"Are there any shared occupations between Maher Sabry and Til Schweiger?"* |
| **Failure Mode** | Over-searching: The model repeatedly queries despite having sufficient information to answer. |
| **Impact** | Increased search cost, unnecessary API calls, and longer latency with no gain in answer quality. |
| **Insight** | Reasoning over retrieved evidence should be prioritized once coverage is sufficient; redundant querying should be suppressed. |

Table 12: Comparison between a successful multi-hop reasoning trajectory and a failure case due to over-searching. The successful case stops searching after gathering sufficient evidence, while the failure case redundantly queries instead of reasoning.

| Step | Successful reasoning | Over-searching failure |
|---|---|---|
| **Input Query** | *Navarone Garibaldi is the half-brother of a singer who has been married how many times?* | *What occupations are shared by Maher Sabry and Til Schweiger?* |
| **Turn 1** | <THINK> Who is Navarone Garibaldi's half-sister? 
 <SEARCH> → Lisa Marie Presley 
 <RECORD> Lisa Marie Presley | <THINK> What is Maher Sabry's occupation? 
 <SEARCH> → Maher Sabry: [occupations] 
 <RECORD> theater director, screenwriter, etc. |
| **Turn 2** | <THINK> How many times has she been married? 
 <SEARCH> → [four marriages] 
 <RECORD> four | <THINK> What is Til Schweiger's occupation? 
 <SEARCH> → Til Schweiger: [occupations] 
 <RECORD> actor, producer, etc. |
| **Turn 3** | <FINAL> The answer is: four. 
 Search terminated successfully. | <THINK> Are there shared occupations? 
 <SEARCH> Maher Sabry (again) 
 <RECORD> (repetition) |
| **Failure Mode** | N/A – correct answer produced with minimal hops. | Over-searching: fails to reason over retrieved evidence, keeps querying. |
| **Insight** | Success relies on using retrieved facts to trigger answer generation. | The model needs an early stop or a reasoning trigger mechanism. |

