# OpenReview forum: "Iterative Self-Incentivization Empowers Large Language Models as Agentic Searchers"
_NeurIPS.cc/2025/Conference — NeurIPS 2025 poster_

### Official Review · Reviewer_XR4h · 2025-06-10

**Clarity:** 3
**Significance:** 2
**Originality:** 2
**Rating:** 3
**Confidence:** 5

**Summary:**

This paper explores the emerging and promising area of agentic search, where language agents interact with search environments through multiple rounds of decision-making. The proposed method, ExSearch, formulates the search process as an iterative EM-based optimization, leveraging trajectory sampling and self-incentivized refinement.

**Questions:**

see above

**Ethical Concerns:**

["NO or VERY MINOR ethics concerns only"]

**Final Justification:**

After reading the author’s response, some of my concerns have been addressed. However, I still remain uncertain about the central focus of the paper, as a substantial portion is devoted to explaining the EM optimization steps, despite the authors’ claim that “the novelty lies more in the agentic search process and its self-improving loop.” Given these considerations, I believe my previous assessment remains a fair evaluation.

**Limitations:**

yes

**Quality:**

3

**Strengths And Weaknesses:**

Strengths:

- The topic of agentic search is timely and important. The paper attempts to move beyond static or single-pass search paradigms by allowing agents to explore search trajectories iteratively.
- The use of EM formulation provides a principled perspective on optimizing the latent trajectories, and the effort to connect it to self-incentivized learning is appreciated.

Weakness:

- The core idea of ExSearch as a multi-round interaction is essentially a workflow definition for iterative search. However, the approach lacks the adaptiveness and flexibility typically expected in agentic behavior. Compared with methods like Search-R1, which perform action planning in a test-time reasoning loop, ExSearch appears relatively rigid and less autonomous in decision-making.
- While the EM-based optimization is well-written, it essentially reduces to importance sampling over trajectories, followed by distribution fitting. This is conceptually similar to ideas in actor-critic RL and related methods, where EM-like steps are implicitly embedded. Thus, although the derivation is mathematically sound, it does not offer a compelling methodological innovation.
- A critical flaw in the experimental design lies in the lack of consistency across backbone LLMs. Many baselines in Table 1 use different language models, making direct comparison problematic. For a fair and scientific evaluation, it is crucial to control such variables, including the retriever, top-k settings etc. The current setup risks conflating performance gains with model differences.
- The datasets used (e.g., HotpotQA) are relatively old and may have already been seen during LLM pretraining, weakening the evidence for generalization. Given the agentic search setting, it would be more convincing to include challenging and realistic benchmarks like GAIA or WebWalker, which require true multi-step reasoning and adaptive behavior.

Minor typos:

line 112: $t$th -> $t$-th

line 160: a blank square in the right side

---

> ### Author Rebuttal · Authors · 2025-07-31
>
> Dear Reviewer,
>
> Thank you for your recognition of this work. We truly appreciate your thoughtful reviews and suggestions, which have helped us improve our work. In response to your feedback, we have made the following revisions:
> 1. Explained the flexibility and extendability of ExSearch (addressing Weakness 1).
> 2. Clarified the role of the Importance sample and the core contribution (addressing Weakness 2).
> 3. Clarified the experiment setup (addressing Weakness 3).
> 4. Added new experiments on GAIA and WebWalker datasets (addressing Weakness 4).
>
>
> ---
>
> # Response to Weakness 1: ExSearch is Flexible and Extensible
>
> We respectfully point out that ExSearch provides a foundational and flexible framework for the model to learn to retrieve useful information through self-incentivization. The current version of our workflow includes, but is not limited to, the Search-R1 [1] workflow, and is designed to be extensible.
>
> Specifically, Search-R1 iteratively formulates new queries based on previously retrieved documents. This capability is already integrated into our method, where the LLMs continuously reflect on what additional information needs to be retrieved and generate a query accordingly. Furthermore, our actions also include reflecting on the quality of documents and ranking the retrieved documents. Both are key primitive actions fully compatible with the ExSearch action planning loop. Our ablation experiments in Section 5.2 further demonstrate the necessity of these actions.
>
> Additionally, we would like to note that Search-R1 was officially released on March 12, making it a concurrent work with ours according to NeurIPS' review guidelines [2]. In summary, we believe our workflow is compatible with Search-R1, and we present a novel and extensible framework.
>
> $    $
>
> # Response to Weakness 2: Clarify the Role of Importance Sample
>
> Thank you for the feedback. Importance sampling [3,4] is a basic and essential technique used in various optimization frameworks, like reinforcement learning or probabilistic statistical frameworks. We would like to clarify that the primary role of importance sampling in our work is to simplify the computation of the search trajectory likelihood, rather than serving as the core contribution.
>
> The key contributions of this work lie in the **ExSearch exploratory search framework**, which includes:
> 1. An exploratory search framework that empowers LLMs to engage in agentic retrieval through a self-incentivization process. This allows the model to iteratively refine its search strategy, selecting and reasoning over the most relevant information.
> 2. A theoretical foundation for this framework, which provides guarantees for convergence and demonstrates how our approach can be extended to more general scenarios, enhancing the flexibility and scalability of the method.
>
> > Besides, **the training algorithm in ExSearch diverges from actor-critic by**:
> > 1. Latent Variable Modeling: Latent variable optimization and reinforcement learning optimization are two distinct paradigms. Our ExSearch models search behavior as latent variables and uses an iterative process with maximum likelihood estimation to infer the distribution of latent variables and optimal parameters. Actor-Critic, in contrast, focuses on policy optimization in reinforcement learning, where the goal is to maximize cumulative rewards.
> > 2. Reward Calculation: Our approach calculates the reward from the log-likelihood of the generated answer, which emphasizes probabilistic inference instead of rewards provided by a critic model or the environment.
>
>
> We believe the novelty lies more in the agentic search process and its self-improving loop (which has been recognized by reviewers zB4g, RVTA, and ymBc), rather than in the specific application of importance sampling. We will clarify these points in the revised version of the paper to better highlight the methodological innovations in our approach.
>
> $    $
>
> # Response to Weakness 3: Our Experiments Include Both Open-Source and Closed-Source Models Under the Same Retriever
>
> Thanks for this valuable suggestion. We agree that keeping experimental settings consistent across baselines is crucial for fair comparison, and this principle guided our design. We believe there might have been a misunderstanding.
>
> ### **1. Consistent Experimental Setup**
> As described in Section 4 (Line 185), all methods in Table 1 use the same retriever, ColBERTv2.0. Furthermore, Line 178 specifies that the number of retrieved documents is fixed across baselines: we follow common practice by using 10 documents for advanced RAG baselines and 5 for iterative retrieval RAG baselines (the same counts used for ExSearch).
>
> ### **2. LLM Backbones**
> RAG research includes many baselines; some rely on closed-source models or don't release code, as explained in Line 181 of our paper. Therefore, our primary focus is on using smaller, open-source backbones and comparing against baselines with equivalent or even stronger backbones. Here are two examples.
>
> 1. Both Search-R1 and ExSearch use Qwen-2.5-7B-instruct, yet ExSearch consistently performs better, as shown below.
>
> >| Method    | NQ (F1 / EM / ACC) | HotpotQA (F1 / EM / ACC) | MuSiQue (F1 / EM / ACC) | 2WikiQA (F1 / EM / ACC) |
> >|:-----------|:--------------------:|:--------------------------:|:-------------------------:|:-------------------------:|
> >| Search-R1 | 54.26 / 42.21 / 51.35 | 58.04 / 46.51 / 52.46 | 30.03 / 21.21 / 23.37 | 52.62 / 49.64 / 50.43 |
> >| ExSearch  | 56.37 / 47.07 / 51.75 | 62.59 / 50.35 / 54.32 | 29.68 / 22.03 / 24.34 | 57.14 / 52.62 / 54.37 |
> >
> > **Caption:** The results from Table 1 in our submitted paper.  All these methods use `Qwen-2.5-7B-instruct` as the backbone. EM refers to the exactly match;  ACC refers to the accuracy.
> >
> >| Method        | NQ (F1 / EM / ACC) | HotpotQA (F1 / EM / ACC) | MuSiQue (F1 / EM / ACC) | 2WikiQA (F1 / EM / ACC) |
> >|:---------------|:--------------------:|:--------------------------:|:-------------------------:|:-------------------------:|
> >| InstructRAG   | 39.21 / 37.82 / 37.58 | 37.31 / 36.77 / 35.31 | 25.88 / 14.94 / 20.45 | 40.01 / 44.57 / 38.91 |
> >| RAG-DDR       | 40.74 / 28.76 / 30.51 | 31.71 / 40.04 / 42.41 | 13.54 / 10.57 / 14.21 | 38.40 / 35.44 / 37.41 |
> >| ExSearch      | 55.21 / 43.71 / 50.76 | 60.72 / 47.59 / 53.59 | 30.83 / 20.98 / 24.65 | – / – / –              |
> >
> >**Caption:** The results from Table 1 in our submitted paper.  All these methods use `Llama-3.1-8B-instruct` as the backbone. EM refers to the exactly match;  ACC refers to the accuracy.
>
> 2. While DSPy is officially implemented only with the powerful GPT-3.5, ExSearch still surpasses its performance using the smaller, open-source Qwen-2.5-7B-instruct.
>
> Besides the above two experimental comparisons, Table 1 includes more comparisons between ExSearch and previous baselines using the same backbone LLMs under the same retrieval setting. The experiment as well as the experimental results of this work have also been acknowledged by reviewers zB4g, RVTA, and ymBc, showing the effectiveness of our method.
>
> We will further clarify these experimental settings in the revised paper to ensure no ambiguity.
>
>
> $     $
>
> # Response to Weakness 4: Why Existing Datasets and Newly Added Experiment
>
> Thank you for the suggestion. Below, we explain why we use existing datasets and report results from additional experiments following your suggestion.
>
> ### **1. Reason for Our Experimental Benchmarks:**
>
> We chose datasets such as HotpotQA to maintain consistency with prior baselines. The **base models, experimental setup, and datasets** strictly follow well-known baselines and previous work. This consistency enables direct, fair comparison and shows that ExSearch outperforms existing methods under the same conditions.
>
> ### **2. New Experiments on GAIA and WebWalker:**
>
> We agree that more challenging benchmarks strengthen validation. Following your suggestion, we additionally evaluate on WebWalker and GAIA, two more realistic, multi-step benchmarks. The results are reported as follows.
>
>
> | Method  | GAIA (Accuracy %) | WebWalker (Accuracy %) |
> |:-----------------------|:---------:|:-----:|
> | Search-R1 + GRPO   |  23.30  | 22.06    |
> | Search-R1 + PPO     | 25.24 | 23.24 |
> | ExSearch | 27.18 | 25.59 |
> | ExSearch (w/ re-ranking in Section 6) | 28.16  | 26.03 |
>
> **Caption:** We use the official metric, i.e., accuracy, from GAIA and WebWalker, respectively. We use the Qwen-2.5-7B-instruct as the backbone LLMs for all methods and use the Google search as the retriever.
>
> Our results show that ExSearch achieves an accuracy of 25.59  in the WebWalker dataset, while the baseline, e.g., Search-R1, only reaches 23.24  with a statistically significant difference (two-tailed t-test under $\alpha=0.05$). This demonstrates the advantage of our method in a more realistic and complex benchmark that requires multi-step reasoning and adaptive behavior.
>
> We will include these results in the revised version of the paper, along with additional analysis on WebWalker, to further validate the advantage of our method on more challenging datasets. We hope this additional experimental evidence addresses your concern and strengthens the case for the effectiveness of ExSearch.
>
>
> ---
>
> Once again, thank you for your review! We will update the final version of our paper with all the revisions discussed. We hope our responses address your concerns, and we are happy to answer any further questions.
>
>
> $     $
>
> $     $
>
> **Reference**
>
> [1]  Jin B, Zeng H, Yue Z, et al. Search-r1: Training LLMs to reason and leverage search engines with reinforcement learning[J]. arXiv preprint arXiv:2503.09516, 2025.
>
> [2] https://neurips.cc/Conferences/2025/CallForPapers
>
> [3] Ishop C M, Nasrabadi N M. Pattern recognition and machine learning[M]. New York: Springer, 2006.
>
> [4] Surya T Tokdar and Robert E Kass. Importance sampling: a review. Wiley Interdisciplinary
> Reviews: Computational Statistics, 2, 2010.

---

### Official Review · Reviewer_ymBc · 2025-06-21

**Clarity:** 2
**Significance:** 3
**Originality:** 3
**Rating:** 5
**Confidence:** 5

**Summary:**

This paper presents a theoretical framework for training iterative retrieval-and-generation agents through the lens of the Expectation-Maximization (EM) algorithm. The proposed ExSearch algorithm casts the E-step as the agent generating multiple search queries and receiving rewards based on the quality of retrieved information. The M-step then applies importance sampling over the search trajectories to compute a loss used to update the parameters of the language model. This EM-inspired formulation provides a principled approach to learning from the agent’s interaction with the retrieval environment.

**Questions:**

1. In the EM-style training algorithm illustrated in Figure 2, the key novelty over previous approaches such as Search-R1 appears to lie in the E-step, where multiple trajectories are sampled, and reward-based weights are used to reweight the updates during training. However, this seems conceptually similar to Group Relative Policy Optimization (GRPO), where a batch of trajectories is used to compute group-level advantages for policy updates. Could the authors clarify the conceptual and algorithmic distinctions between their method and GRPO?

2. As raised in the earlier weaknesses, the term $p(z|x;\theta)$ represents the model’s distribution over search trajectories $z$. However, it remains unclear how the reward associated with each $z$ is quantified during training. Is there any form of ground-truth reasoning path used to evaluate the quality of $z$, or is the reward computed via another proxy, such as perplexity or task performance? More details on how this reward signal is defined and grounded would be helpful.

3. Lines 270–271 refer to "1,00" samples, which seems to be a typographical error. Presumably this is meant to be 1,000 samples used for warm-up. Based on Figure 4, the performance gain from using 1,000 versus 800 samples appears marginal. Could the authors clarify why 1,000 was selected as the final choice, and whether there is a principled or empirical reason behind this threshold?

4. Regarding Table 1, the visual design raises several concerns. Both purple and green are used to highlight "Ours" methods, yet the distinction between them is unclear. Moreover, in the green-shaded region, some models outperform the purple-highlighted baseline but are not bolded, which is confusing. Could the authors clarify the meaning behind these color codings and whether the comparison settings (e.g., model size, training data) are strictly comparable across the variants in the table?

**Ethical Concerns:**

["NO or VERY MINOR ethics concerns only"]

**Final Justification:**

I have read the authors’ response and appreciate the clarifications provided. The rebuttal addresses some of my concerns and offers additional detail, particularly regarding the technical formulation. I'll raise the score.

**Limitations:**

yes

**Quality:**

4

**Strengths And Weaknesses:**

# Strengths

* The proposed framework is elegant and intuitive, offering a principled formulation for iterative retrieval and generation. Its effectiveness is supported by thorough comparisons with a wide range of baselines, including standard LLMs, retrieval-augmented generation (RAG), and existing agentic search methods.

# Weaknesses:

* While the EM-based formulation provides a theoretical foundation, the paper introduces considerable mathematical complexity, e.g., ELBO derivations and convergence analysis, for what is essentially a straightforward multi-step retrieval and generation process. This added formalism may reduce the readability and accessibility of the method.

* Despite the mathematical rigor, the method section lacks sufficient clarity on practical implementation details. For example, it is unclear how key components in the loss functions $L_R$ and $L_A$ are instantiated: such as how $w(z)$ is defined, or how $p(y|x, z; \theta)$ and $p(z|x; \theta)$ are computed to optimize the LLM parameters. These ambiguities hinder reproducibility and may leave readers uncertain about the concrete mechanics of the proposed approach.

---

> ### Author Rebuttal · Authors · 2025-07-30
>
> Dear Reviewer,
>
> Thank you for your recognition of this work. We truly appreciate your thoughtful reviews and suggestions, which have helped us improve our work. In response to your feedback, we have made the following revisions:
> 1. Simplified mathematical derivations (addressing Weakness 1).
> 2. Provided a clearer explanation of the key components in our loss function (addressing Weakness 2 and Question 2).
> 3. Compared our method with GRPO (addressing Question 1).
> 4. Clarified the scale of warm-up training data (addressing Question 3).
> 5. Explained the color coding in Table 1 (addressing Question 4).
>
> ---
>
> $     $
>
> # Response to Weakness 1: Theoretical Analysis
>
> We fully understand the concern that the mathematical derivation and analysis may introduce complexity, and **we plan to strike a better balance**.
>
> **Why do we provide a theoretical analysis?** The introduction of the Generalized EM (GEM) framework is essential for ensuring the theoretical soundness and convergence guarantees of ExSearch (as also recognized by reviewer RVTA). We aim to provide an explainable foundation for our method (e.g., how & why it works) for readers and distinguish it from simpler multi-step retrieval processes (particularly those based on prompt engineering). We hope this ensures the long-term convergence and performance across different retrieval scenarios of our method.
>
> **We will further simplify the mathematical derivation.** We agree with your suggestion. In future iterations of the paper, we plan to simplify some of the mathematical details and present the key ideas in a more intuitive and accessible way. This will include providing clearer preliminary explanations and potentially reformatting the formalism to focus on the core insights without losing rigor. We also plan to streamline the ELBO derivations and convergence analysis to make them more digestible while maintaining theoretical guarantees.
>
> We hope these changes will address your concerns.
>
> $       $
>
> # Response to Weakness 2 and Question 2: Further Illustrating the Loss Function and Trajectory Weight
>
> We appreciate the comment and clarify below how each quantity is instantiated in practice (see Eq. (6) in the paper).
>
> 1. **Probability of a search trajectory $p(z \mid x)$**: A trajectory $z=$ {$(x_i, d_i, e_i) \mid i\in [|z|]$} consists of the *think* (query $x_i$), *search* (retrieved document $d_i$) and *record* (evidence $e_i$) actions.
>
> $      $ $      $ $      $ $      $ $      $ $      $ Thus, $p(z \mid x; \theta) = \prod_{i=1}^{|z|} p(x_i \mid x, z_{<i}; \theta)\; p(e_i \mid x_i, d_i; \theta)$, where both factors are standard token-level LM likelihoods (sum of log-probs over generated tokens) computed by the current model $\theta$.
>
> 2. **Answer likelihood conditioned on the trajectory $p(y \mid x; z)$**:
> We compute the probability of the model in generating the ground-truth answer y:
> $p(y \mid x, z; \theta) = \prod_{t=1}^{|y|} p(y_t \mid x, z, y_{<t}; \theta)$,
> which is the LLM token log-likelihood under $\theta$ with the entire trajectory $z$ in context.
>
> 3. **Trajectory importance weight $w(z)$**: Using importance sampling (Eq. (5)), the unnormalized weight is $\tilde{w}(z) \propto p(y \mid x, z; \theta_t)$, and we normalize over the sampled trajectories $w(z) = \frac{\\tilde{w}(z)}{\\sum_{j=1}^{M} \\tilde{w}(z^{j})}$.
>
> 4. **Loss instantiation**: Given sampled trajectories $z$, weights $w(z)$, and the LLM parameter $\theta$, we have:
> - Reasoning loss $L_{\mathcal{R}} = - \sum_{m} w(z) \Big( \sum_i \log p(x_i \mid x, z_{<i};\theta ) + \log p(e_i \mid x_i, d_i;\theta) \Big)$, as introduced in Line 88 and Line 129 of the submitted paper;
> - Answer loss $L_{\mathcal{A}} = - \sum_{m} w(z) \log p(y \mid x, z; \theta)$, as introduced in Line 131 of the submitted paper;
>
> $      $ $      $ $      $ $      $ $      $ $      $ The final objective is $L = L_\mathcal{R} + L_\mathcal{A}$. We backpropagate through the LLM as usual to update the model parameters.
>
> > Mini example (conceptual)
> > - Input $x$: “Who wrote *The Old Man and the Sea* and in which year did it win the Pulitzer Prize?”
> > - Two sampled trajectories $z^{1}$, $z^{2}$ differ in sub-queries and evidence snippets.
> > - Compute token-level log-likelihoods for each $x_i, e_i$, and for the final answer $y$.
> > - Normalize $\tilde{w}(z^{1}), \tilde{w}(z^{2})$ to get $w(z^{1}), w(z^{2})$.
> > - We sum the weighted losses $L_\mathcal{R}$ and $L_\mathcal{A}$, and update $\theta$.
>
>
> $      $
>
> # Response to Question 1: Discussing Latent Variable Modeling with GRPO
>
> The key differences between our method and previous work, such as Search-R1, are as follows, with more details provided in Appendix C.1:
>
> 1. **Reward Calculation**: Our approach calculates the reward from the log-likelihood of the generated answer, rather than relying on an exact match score between the predicted answer and the ground truth.
> 2. **Trajectory-level Reward Supervision vs. Outcome Reward Supervision**: As shown in Eq. 10 and Appendix Line 743, our weight w(z) evaluates the entire search trajectory’s contribution to generating the answer, rather than focusing solely on the final outcome. The results in the main experiment (Table 1) and the analysis experiments in D.3.4 (Line 919) demonstrate that this more global supervision signal yields better performance.
> 3. **Latent Variable Modeling vs. Reinforcement Learning**: Latent variable optimization and reinforcement learning optimization are two distinct paradigms. Our ExSearch models search behavior as latent variables and use an iterative process with maximum likelihood estimation to infer the distribution of latent variables and optimal parameters. In contrast, GRPO focuses on online policy optimization, updating model parameters via policy gradients.
>
> We will incorporate these clarifications into the revised version of the paper.
>
>
>
> $   $
>
> # Response to Question 3: Explanation about Warmup Training Data Scale
>
> We appreciate this thoughtful review. Below, we respectfully clarify any misunderstanding about our warm-up training data scale and analysis in Section 5.5.
>
> **First, the choice of 1,000 samples in this experiment was primarily driven by cost considerations.** Training with too many samples increases costs and can reduce the scalability of the method. `Therefore, we initially selected 1,000 samples for training (at a cost of $20) and conducted extensive analysis to compare model performance across different initial training data sizes.`
>
> **Second, it is indeed 100 instead of a typographical error.** In Section 5.5 (the paragraph containing lines 270–271), we aim to investigate the impact of initial warm-up on final training performance and empirically demonstrate the minimal data scale to enable the proposed self-training. Thus, we gradually reduced the number of warm-up training samples and compared the final training performance. We found that the model initially trained on 1,000 samples achieved an exact match of 50.35, while no warm-up only led to around 45. However, `the model trained on only 100 samples achieved around 48, outperforming strong baselines and striking an empirical balance between data creation cost and performance.` Thus, we highlight 100 examples in Lines 270–271.
>
> Overall, the focus of this paper is on constructing a foundational agentic search framework rather than optimizing the warm-up training scale. Thus, we initially set the data scale to 1,000 and then gradually reduced the scale for fine-grained validation and comparison. In the revised version of our paper, we will make room to add this content for explanation.
>
>
> $   $
>
> # Response to Question 4: Why Use Two Color Codings
>
> We appreciate this feedback regarding the visual design and clarity of Table 1. The distinctions between the purple and green color codings are intended to highlight different types of experiments, and we will clarify these in the revised version of the paper.
>
> Specifically, due to space constraints in the submitted paper, we combined both the main experimental results and the analysis experiments into a single table (Table 1). To clearly differentiate between the two types of experiments, we use different color codings:
>
> 1. **Upper Section with Green (Main Experiment)**: The upper half of the table presents the performance comparison between our method and previous baselines, which we refer to as the main experiments. In this section, we highlight the results for "Ours" in purple to indicate the primary comparison.
>
> 2. **Lower Section with Green (Analysis Experiment)**: The lower half of the table showcases the analysis experiments to demonstrate the generalizability of our method when applied to different backbone LLMs. These experiments are intended to show that our approach works effectively across multiple model families with varying parameters. To differentiate this section from the main experiment, we highlight "Ours" in green.
>
> In the revised version, we will make room to separate Table 1 into two distinct tables, providing a clearer presentation of our experimental results.
>
>
>
> $   $
>
> ---
>
> Once again, we really appreciate your review! We will update the final version of our paper with all the revisions discussed. We hope our responses address your concerns, and we would be happy to provide further details on any specific aspect if needed.
>
> $   $
>
> **Reference**
>
> [1] Jin B, Zeng H, Yue Z, et al. Search-r1: Training LLMs to reason and leverage search engines with reinforcement learning[J]. arXiv preprint arXiv:2503.09516, 2025.

---

> ### Author Response · Authors · 2025-08-08
> **Thank you for your recognition**
>
> Dear Reviewer ymBc,
>
> Thank you for your recognition of our work. We sincerely appreciate your thoughtful suggestions and kind response, which have been invaluable in improving our paper. We ensure that all the points discussed in the rebuttal, such as simplifying the mathematical derivation and providing more details on warm-up training, will be incorporated into the revised version.
>
> Thank you again for your engagement. We are also happy to provide any further details if needed.
>
> Best wishes,
>
> Authors

---

### Official Review · Reviewer_RVTA · 2025-07-01

**Clarity:** 3
**Significance:** 3
**Originality:** 3
**Rating:** 4
**Confidence:** 4

**Summary:**

This paper introduces a novel framework, EXSEARCH, designed to enhance the capabilities of large language models (LLMs) as information retrieval agents through an iterative self-motivated process. The core of this framework enables LLMs to dynamically perform document retrieval, evidence extraction, and answer aggregation, thereby improving their knowledge acquisition in complex tasks via a self-motivated mechanism.

Main Contributions:

Proposal of the EXSEARCH Framework: EXSEARCH defines three core actions—“thinking” (query generation), “searching” (triggering the retriever), and “recording” (extracting fine-grained evidence)—allowing LLMs to iteratively explore search trajectories and generate final answers based on the entire trajectory.
Theoretical Analysis and Convergence Guarantee: The authors provide a theoretical analysis of the training process, demonstrating that the self-motivated framework achieves stable convergence.
Extensive Experimental Validation: Experiments conducted on multiple knowledge-intensive benchmark datasets show that EXSEARCH significantly outperforms existing retrieval-augmented language models.
Introduction of EXSEARCH-Zoo: This extended resource generalizes EXSEARCH to a broader range of scenarios, including diverse backbone LLMs from different model families and scales, as well as an expanded action space (e.g., document re-ranking).
Methodological Details:

Self-motivated Training Process: EXSEARCH employs a Generalized Expectation-Maximization (GEM) algorithm, alternating between an E-step (trajectory exploration) and an M-step (reweighted trajectory learning). This enables the LLM to learn from its self-generated data, progressively improving its search and reasoning abilities. In the E-step, the LLM generates multiple search trajectories for each input task and assigns importance weights to them; in the M-step, the LLM updates its parameters based on these trajectories and their associated weights.

Experimental Results: EXSEARCH demonstrates substantial improvements over baseline models across several benchmarks. For example, on the HotpotQA dataset, using the 7B Qwen2.5 model, the F1 score increases from 55.40 (RankRAG-70B) and 53.31 (Search-o1-32B) to 62.59. Additionally, EXSEARCH exhibits superior retrieval performance, with recall rates significantly higher than those achieved by using only retrievers or traditional re-ranking methods.

**Questions:**

1. Currently, EXSEARCH performs retrieval at every reasoning step, regardless of necessity. It may be beneficial to develop a more adaptive strategy that enables the model to determine, based on contextual information, when retrieval is required.

**Ethical Concerns:**

["NO or VERY MINOR ethics concerns only"]

**Final Justification:**

Overall, the approach proposed by EXSEARCH is quite effective. I have no further concerns and rate it a 4.

**Limitations:**

yes

**Quality:**

3

**Strengths And Weaknesses:**

Strengths:

1. Self-motivated Learning Mechanism: EXSEARCH adopts a novel self-motivated process, enabling LLMs to iteratively learn to retrieve useful information during reasoning. This internal, adaptive feedback loop guides the learning process without requiring explicit, manually designed intermediate rewards for each agent interaction step.

2. Unique Use of the Generalized EM Algorithm: During the E-step, the LLM generates multiple search trajectories and assigns importance weights to each; in the M-step, it trains using a reweighted loss function based on these trajectories. This iterative process establishes a robust self-motivated loop, allowing the LLM to learn directly from its self-generated and implicitly evaluated data, thereby progressively enhancing its search and reasoning capabilities.

3. Theoretical Analysis and Convergence Guarantee: A significant contribution of the framework is the inclusion of theoretical analysis, establishing convergence guarantees for the training process. This provides a solid mathematical foundation for the framework’s stability, which is often challenging to prove in complex RL-based systems.

Weaknesses:

1. The description of EXSEARCH lacks details regarding its initial training phase and how it addresses the “cold start” problem.
2. Although theoretically sound, implementing the Generalized EM algorithm in large language model training is more complex than methods such as policy gradient (e.g., GRPO). While the self-motivated training process is theoretically guaranteed to converge, in practice, training stability and efficiency may be affected by data quality and the underlying model.
3. EXSEARCH addresses irrelevant information by weighting useful information through the EM process; however, more explicit technical solutions are needed to prevent retrieved content from biasing the training strategy. It remains challenging to ensure that the EM process can fully and implicitly handle this issue.

---

> ### Author Rebuttal · Authors · 2025-07-30
>
> Dear Reviewer,
>
> Thank you for your recognition of this work. We truly appreciate your thoughtful reviews and suggestions, which help us improve our work. In response to your review, we have made the following revisions:
> 1. Clarified details regarding the cold start process (addressing Weakness 1).
> 2. Explained the efficiency and superiority of ExSearch (addressing Weakness 2).
> 3. Discussed strategies to prevent noise during the retrieval process (addressing Weakness 3).
> 4. Addressed enabling adaptive retrieval (addressing Question 1).
>
> ---
>
> $  $
>
> # Response to Weakness 1: Explanation about the Initial Training (Cold Start)
>
> We appreciate your valuable feedback. We've added the following details to our paper for better clarification.
>
> As described in Line 186 (implementation details), to address the cold start problem, we initially fine-tune the model with 1000 pseudo-examples annotated by GPT-4o. This aims to initialize the model with the basic skills required to follow ExSearch's iterative think-search-record pattern.
>
> Specifically, we use GPT-4o to label data that adheres to our agentic framework, ensuring that the generated examples align with the goals of the iterative reasoning process. This process develops the model’s basic capacity to engage in the think-search-record cycle proposed in the ExSearch workflow. These data synthesis details, along with further explanations, have been provided in the ***Data Collection for Warm-Up*** section (Line 1014).
>
> $  $
>
> # Response to Weakness 2: Clarification about Why ExSearch is Efficient
>
> Thank you for your thoughtful comment. We fully agree that implementing the Generalized EM (GEM) algorithm for training LLMs presents complexities, particularly regarding training stability and efficiency. However, we would like to emphasize that `we have simplified ExSearch's training into an elegant self-training manner (which has been recognized via reviewers ymBc and zB4g) and have conducted extensive experiments to demonstrate its practical stability and efficiency.`
>
> We provide the following detailed explanations:
> 1. **Simple Iterative Training Process**: We have simplified the training approach through an iterative process, which helps mitigate sampling instability during training. This process, as discussed in the paper, allows for a more stable and controlled learning environment, ensuring consistent progress without the typical challenges associated with sampling-based methods.
> 2. **Empirical Validation of Stability and Efficiency**: To validate the stability and efficiency of ExSearch, we conduct experiments on multiple datasets. Across these datasets, we observed consistent, stable improvements in model performance. This supports our claim that the proposed training method is stable across various tasks and data settings.
> 3. **Model Variety**: As illustrated in Figure 1 and Table 1 in our submitted paper, we evaluated ExSearch across a variety of LLMs, including smaller and larger models, and observed stable improvements in performance in all cases. This further reinforces that ExSearch scales well and is effective across different architectures.
> 4. **Efficiency Comparison**: In terms of training efficiency, we conducted a direct comparison between ExSearch and other leading methods such as Search-R1 (trained via PPO / GRPO), as reported below.
> > | Method | Training Hours | Performance (Accuracy) |
> > |:--|:--:|:--:|
> > | ExSearch           |    8.2       | 54.32 |
> > | Search-R1 (PPO)    |  19.8      | 46.51 |
> > | Search-R1 (GRPO)   |  16.4    | 46.12 |
> >
> >**Caption**: The count of training hours of different methods on the HotpotQA dataset.
>
> $  $ $  $ $  $ $  $ $  $ We find that ExSearch achieves better performance with fewer training hours.
>
> $  $ $  $ $  $ $  $ $  $ We also report average token consumption and the number of retriever calls per example (e.g., per input query).
>
> >|       Method        | Avg. Input Token Consumption | Avg. Output Token Consumption | Retriever Calls | Performance (Accuracy) |
> >|:-------------------|:---------------------------:|:----------------------------:|:---------------:|:-----------:|
> >| ExSearch            | 1859.98                    | 325.30                      | 2.45            | 54.32       |
> >| Search‑R1 (PPO)     | 2059.98                    | 304.21                      | 3.14            | 46.51       |
> >| Search‑R1 (GRPO)    | 2351.73                    | 332.12                      | 3.56            | 46.12       |
> >
> >**Caption**: The statistics of different methods during the inference on the HotpotQA dataset.
>
> $  $ $  $ $  $ $  $ $  $ We observe that ExSearch achieves higher accuracy while consuming fewer tokens and making fewer retriever calls, indicating greater efficiency in both training and inference.
>
> We believe that these explanations address the concerns regarding the practical implementation of GEM in LLM training, confirming that ExSearch offers a robust and efficient solution for dynamic retrieval and reasoning tasks.
>
> $  $
>
> # Response to Weakness 3: Preventing the Noise in ExSearch
>
> We appreciate this insightful suggestion. Addressing irrelevant information is a long-standing research challenge in Information Retrieval (IR), involving efforts from many research directions, e.g., document re-ranking or content extraction. In our work, we have considered this by extending the document selection action in ExSearch. As described in Section 7 of our submitted paper, we incorporate a re-ranking step into the reasoning process to address this concern.
>
> Specifically, in Section 7.1, we extend the original ExSearch workflow from the pattern `think -> search -> record` to `think -> search -> re-rank -> record`. The added re-rank step allows us to re-rank the retrieved documents based on their relevance to the current sub-query. Only the top-k documents, as determined by this ranking process, are then passed into the reasoning trajectory. This approach ensures that only the most relevant and supportive information contributes to the learning process, effectively mitigating the risk of irrelevant content biasing the training strategy. This modification remains fully compatible with the original ExSearch training framework, as it only changes the sampling method. By adding this ranking step, we provide an elegant solution that reduces the bias from irrelevant retrieved information, leading to about a 2.8% to 5.0% point improvement in performance. This change enhances the stability of the training process while preserving the core dynamics of ExSearch.
>
> Debiasing irrelevant information presents a non-trivial challenge in IR. We are actively exploring more advanced technical solutions. Future iterations of ExSearch will include more useful solutions, such as dynamic filtering mechanisms, to better control and reduce the impact of irrelevant content on the model's training.
>
> $  $
>
> # Response to Question 1: Enabling Adaptive Retrieval in ExSearch
>
> Thank you for your thoughtful suggestion. Handling **when to retrieve** is a long-standing challenge in IR and RAG (e.g., adaptive retrieval, knowledge-boundary reasoning). A key principle there is letting the model explicitly know what it knows and what it still needs. The current version of ExSearch already follows this principle:
> 1. **Think step**: The LLM reasons over the current context and formulates a new query to specify missing information.
> 2. **Record step**: It reflects on the usefulness of retrieved documents by extracting key evidence, effectively filtering noise.
>
> Beyond the current version, we agree that integrating additional mechanisms is beneficial. Inspired by adaptive retrieval/reasoning work, we provide the following technical solutions:
> 1. Pre-determining the query complexity and then selecting the most fitting retrieval strategies [1, 2]
> 2. More explicit need-to-know reasoning before each retrieval call [3, 4].
> 3. Reward/penalty on unnecessary retrieval to discourage overuse of the search engine [3].
> 4.  Adaptive stopping criteria and budgets for retrieval depth.
>
> These solutions, particularly the *1. query complexity assessment* and *4. Adaptive stopping criteria and budgets for retrieval depth* can be seamlessly integrated into ExSearch, while the other solutions can be plugged into ExSearch's action loop.
>
> We will add these discussions to the revised paper to address your concern more concretely.
>
> ---
>
>
> Once again, we really appreciate your review! We will update the final version of our paper with all the revisions discussed. We hope our responses address your concerns, and we would be happy to provide further details on any specific aspect if needed.
>
> $     $
>
> **Reference**
>
> [1] Jeong S, Baek J, Cho S, et al. Adaptive-rag: Learning to adapt retrieval-augmented large language models through question complexity. arXiv 2024[J]. arXiv preprint arXiv:2403.14403.
>
> [2] Mallen A, Asai A, Zhong V, et al. When not to trust language models: Investigating effectiveness of parametric and non-parametric memories[J]. arXiv preprint arXiv:2212.10511, 2022.
>
> [3] Ren B, Qiao S, Yu W, et al. KnowRL: Exploring Knowledgeable Reinforcement Learning for Factuality[J]. arXiv preprint arXiv:2506.19807, 2025.
>
> [4] Qiao S, Qiu Z, Ren B, et al. Agentic knowledgeable self-awareness[J]. arXiv preprint arXiv:2504.03553, 2025.

---

> > ### Comment · Reviewer_RVTA · 2025-08-04
> >
> > Thank you to the authors for their response. I have no further questions and will maintain my borderline accept decision.

---

> ### Author Response · Authors · 2025-08-04
> **Follow-up on Rebuttal**
>
> Dear reviewer RVTA,
>
>
> We truly appreciate your engagement with our work.
>
> To ensure we make the best use of the remaining discussion time, we would like to kindly check if our responses have clarified your main concerns. Please let us know if any part of our response is unclear or if there are any remaining concerns we can address. We would be more than happy to provide any further details or adjustments you might need.
>
> Thank you once again for your thoughtful engagement with our work.
>
> Best wishes,
>
>
>
> Authors

---

### Official Review · Reviewer_zB4g · 2025-07-02

**Clarity:** 3
**Significance:** 3
**Originality:** 3
**Rating:** 5
**Confidence:** 3

**Summary:**

This paper focuses on the design of an LLM-based information retrieval method enhanced with the Expectation-Maximization (EM) algorithm. To accommodate more complex query scenarios, it models multi-turn, autoregressive retrieval reasoning paths—including generated sub-queries, retrieved documents, and accumulated evidence—as latent variables. Using the EM algorithm, it optimizes the expected log-likelihood of the joint distribution $\mathbb{E}_{\text{reasoning path}} \left[  P(\text{reasoning path}, \text{answer} \mid \text{query}) \right]$. Compared to directly optimizing the log-likelihood of the answer given a query, this approach enables more effective exploration of reasoning paths, resulting in more accurate and interpretable answers for challenging information retrieval tasks. This paper also conduct extensive experiments across multiple datasets and foundation LLMs, benchmarking against a diverse set of strong baselines. Detailed ablation studies, in-depth analysis, and case studies further demonstrate the effectiveness of the proposed approach.

**Questions:**

**Question 1:**
 I’m curious about how EXSEARCH compares to other baselines in terms of training and inference efficiency. Some quantitative insights or comparisons in this regard would be helpful for understanding its practical applicability.

**Question 2:**
 Given the discussion in Weakness 2 about the EM algorithm's local convergence properties, it would be valuable to understand how EXSEARCH behaves under different initialization strategies—such as varying prompt templates or parameter initializations. Has the robustness of convergence under these variations been explored?

**Question 3:**
 As noted in Weakness 1, the term *“Iterative Self-Incentivization”* seems to reflect the optimization dynamics introduced by the EM algorithm. At the same time, EXSEARCH could reasonably be seen as a form of “agentic search” within the information retrieval domain. This raises an open question about naming: should the title highlight the underlying method (e.g., EM-enhanced LLM-based IR) or the emergent behavior (e.g., agentic characteristics)? I’d be interested to hear the authors’ perspective on this balance.

**Ethical Concerns:**

["NO or VERY MINOR ethics concerns only"]

**Final Justification:**

The rebuttal has addressed the majority of my concerns, and based on this, I would like to recommend that this paper be accepted.

**Limitations:**

yes

**Quality:**

3

**Strengths And Weaknesses:**

**Strengths:**

* **Strength 1:**  The core idea of this paper is to treat the retrieval reasoning path as an unobserved latent variable in an autoregressive manner and optimize it through iterative self-incentivization using the Expectation-Maximization (EM) algorithm. The integration of EM-based modeling is both conceptually sound and practically elegant—resulting in a streamlined algorithm that is not only effective but also insightful. This framework offers a flexible foundation: by tailoring the latent variable assumptions to specific application scenarios, corresponding EM algorithms can be readily designed and adapted.
* **Strength 2:**  The paper is clearly written and well-organized, with clean and informative figures and tables. The presentation is easy to follow, making the content highly accessible to readers. Furthermore, the appendix provides a thorough discussion of the paper’s limitations, ethical considerations, and societal impacts, along with detailed analyses and commentary on both successful and failure cases.
* **Strength 3:**  The paper conducts extensive experiments across multiple datasets and foundation LLMs, benchmarking against a wide range of strong baselines. Comprehensive ablation studies, in-depth analyses, and carefully examined case studies further validate the effectiveness and robustness of the proposed approach.


**Weaknesses:**

* **Weakness 1:**  The core contribution of this paper lies in integrating the EM algorithm with LLM-based information retrieval. However, the current title—*“Iterative Self-Incentivization Empowers Large Language Models as Agentic Searchers”*—does not reflect either the focus on information retrieval or the role of the EM algorithm. In fact, the phrase *“Agentic Searchers”* initially led me to believe the paper was about autonomous agent architecture design, similar to works like *“Automated Design of Agentic Systems”* [1] and *“AgentSquare: Automatic LLM Agent Search in Modular Design Space”* [2]. Given that the experimental setup primarily benchmarks against information retrieval datasets and RAG baselines—and does not compare with other agent search methods (e.g., [1], [2]) or broader agent scenarios (e.g., embodied agents, game environments, tool use)—emphasizing "information retrieval" in the title would help clarify the paper’s true focus at first glance and better align with the experimental scope.
* **Weakness 2:** The theoretical contribution emphasized in the main text—namely, the convergence of the proposed algorithm—does not appear to leverage any domain-specific properties of information retrieval. To my knowledge, it largely reiterates the standard convergence behavior of the EM algorithm with respect to the value of the loss function, which has already been thoroughly established in classical texts such as PRML [3]. This discussion may be more appropriately placed in the preliminaries section. Moreover, the Pinsker’s inequality derivations in the appendix could benefit from clearer attribution—specifically, which parts are novel contributions of this paper. It is also worth noting that the convergence of the EM algorithm in terms of loss value does not imply convergence to a global optimum; it merely guarantees convergence to a stationary point. Given that loss functions in information retrieval are often ReLU-based and may lack strong convexity, the EM algorithm is unlikely to exhibit global convergence in this context. Exploring the impact of different initialization points might provide a more informative and realistic characterization of the model’s convergence behavior.


**Minor grammatical issues:**

- [1] Line 38: “an LLM” → consider changing to “a LLM”.
- [2] Line 112: “$t$th” → might be written as “$t$-th”.

[1] Hu S, Lu C, Clune J. Automated design of agentic systems[J]. ICLR, 2025.

[2] Shang Y, Li Y, Zhao K, et al. Agentsquare: Automatic llm agent search in modular design space[J]. ICLR, 2025.

[3] Bishop C M, Nasrabadi N M. Pattern recognition and machine learning[M]. New York: springer, 2006.

---

> ### Author Rebuttal · Authors · 2025-07-30
>
> Dear Reviewer,
>
> Thank you for your recognition of this work. We truly appreciate your thoughtful suggestions and guidance, which help us improve our work. In response to your review, we have made the following revisions:
> 1. Clarified key terms in the title, specifically "agentic", and considered more intuitive titles (addressing Weakness 1 and Question 3).
> 2. Reorganized the analysis of convergence and shared more practical implementation insights (addressing Weakness 2 and Question 2).
> 3. Compared the training and inference costs of the proposed EXSEARCH with those of the baselines.
>
> More details are provided below.
>
> ---
>
> $  $
>
> # Response to Weakness 1 and Question 3: Explanation about Our Title
>
> Our use of the phrase "agentic searchers" aligns with the LLM-based agent literature [1, 2]. In this context, **"agentic"** emphasizes an LLM that actively plans, executes, and reflects on a multi-step retrieval process (e.g., think → search → record in this work), rather than a fully autonomous embodied agent.
>
> To avoid ambiguity and explicitly highlight the Information Retrieval (IR) focus, we are willing to adopt a more straightforward title for the camera-ready version, such as *Enabling Large Language Models to Retrieve via Iterative Self-Incentivization*. This title explicitly positions retrieval as the primary application domain and underscores "Self-Incentivization" as the key algorithmic contribution that underpins our Generalized Expectation-Maximization algorithm. We believe this new title will more precisely convey the paper's scope and contributions, making it easier for the research community to understand how our work advances LLM capabilities within information retrieval through a novel learning paradigm.
>
> $   $
>
> # Response to Weakness 2 and Question 2: Explanation about the ExSearch Convergence.
>
> We deeply appreciate your insightful guidance on our theoretical analysis. Below are our detailed responses.
>
> **For the content re-organization**: Our core contribution lies in providing ExSearch, a flexible agentic search framework, complemented by a rigorous theoretical foundation based on the Expectation-Maximization (EM) algorithm (Section 2.2, Appendix D.1 and D.3), which demonstrates its progressive improvement. We agree that relocating corresponding detailed derivations to the Preliminaries section can enhance overall clarity, and we will revise accordingly.
>
> **For global optimality**: Achieving global optimality for neural-based models remains a crucial yet complex challenge. Many advanced methods, including various variational and gradient-based algorithms, often struggle to guarantee global optima. In agreement with your suggestion, the impact of factors such as different parameter initialization should indeed be explored. In this work, we have taken it as a guiding principle to conduct multiple trials with various initialization strategies, including:
> 1. **Exploring different model architectures**: For example, we implement our method with Qwen-2.5-7B-instruct and Llama-3.1-8B, which possess distinct layers, hidden state sizes, and other architectural differences (See Table 1).
> 2. **Utilizing models with diverse initial parameters**: For example, Qwen-2.5-7B-instruct and Llama-3.1-8B are pre-trained on different corpora, leading to varied initial neural parameters (See Table 1).
> 3. **Varying model parameter scales**: We evaluate our method with models ranging in size, such as Qwen-2.5-3B-instruct and Mistral-2501-24B-instruct (See Table 1).
> 4. **Adjusting the quantity of warm-up training data**: We evaluate models that are initially trained on varying amounts of warm-up data, from 100 to 1000 examples (See Section 5.5).
>
> Due to the limited space in our previous submission, we also explored various other settings beyond the experiments reported above. These included: (i) different random seeds, (ii) various prompt templates for warm-up, and (iii) different sampling temperatures for trajectory generation.
>
> For our experiments, the random seeds and temperatures were collected from well-known baselines, such as Search-R1, and a commonly used toolkit, namely Verl [3].
>
> This table shows how the model performs with varying random seeds:
>
> | Random Seed |   F1  |   EM  |  ACC |
> |--|--|--|--|
> | 42  | 62.59 | 50.35 | 54.32|
> | 24 | 61.47 | 50.15 | 54.78|
> | 1 | 62.07 | 51.06 | 54.08|
> | 12345 | 62.57 | 50.94 | 53.75|
>
> **Caption**: Model performance on HotpotQA with different random seeds, using Qwen-2.5-7B-instruct as the backbone LLM (with a temperature of 1.0).
>
> This table shows how the model performs with varying sampling temperatures:
>
> | Sampling Temperature |   F1  |   EM  |  ACC |
> |--|--|--|--|
> | 0.7 | 61.98 | 51.05 | 54.04|
> | 0.8  | 62.59 | 50.35 | 54.32|
> | 0.9  | 62.94 | 50.94 | 54.63|
> | 1.0  | 62.03 | 51.56 | 54.81|
>
> **Caption**: Model performance on HotpotQA with different sampling temperatures, using Qwen-2.5-7B-instruct as the backbone LLM (with a random seed of 42).
>
> We found that these initial explorations did not reveal significant differences in final evaluation performance (two-tailed t-test with $\alpha = 0.05$). We will make more room in our paper and discuss these findings, further enriching the practical insight in understanding ExSearch's optimization landscape.
>
> $  $
>
> # Response to Question 1: Added Training Cost Comparison Between ExSearch and Baselines
>
> Thank you for your suggestion. We have included a quantitative analysis of both training and inference costs below.
>
> For the training cost comparison, we implemented our proposed method, ExSearch, and the baselines (Search-R1) using the same backbone LLM, Qwen-2.5-7B-instruct. All methods utilize ColBERTv2.0 as the retriever, with the number of retrieved documents (top-k) consistently set to 5. The following details are based on experiments conducted by training on the HotpotQA training split and evaluating on the HotpotQA validation split.
>
> 1. **Training Cost:** As reported below, we find that ExSearch achieves better performance with fewer training hours.
>
> | Method | Training Hours | Performance (Accuracy) |
> |:--|:--:|:--:|
> | ExSearch           |    8.2       | 54.32 |
> | Search-R1 (PPO)    |  19.8      | 46.51 |
> | Search-R1 (GRPO)   |  16.4    | 46.12 |
>
> **Caption**: The count of training hours of different methods on the HotpotQA dataset.
>
> 2. **Inference Cost:** We also report average token consumption and the number of retriever calls per example (e.g., per input query).
>
> |       Method        | Avg. Input Token Consumption | Avg. Output Token Consumption | Retriever Calls | Performance (Accuracy) |
> |:-------------------|:---------------------------:|:----------------------------:|:---------------:|:-----------:|
> | ExSearch            | 1859.98                    | 325.30                      | 2.45            | 54.32       |
> | Search‑R1 (PPO)     | 2059.98                    | 304.21                      | 3.14            | 46.51       |
> | Search‑R1 (GRPO)    | 2351.73                    | 332.12                      | 3.56            | 46.12       |
>
> **Caption**: The statistics of different methods during the inference on the HotpotQA dataset.
>
> We also observe higher accuracy of ExSearch while consuming fewer tokens and making fewer retriever calls, indicating greater efficiency.
>
> ---
>
> Once again, thank you for your review! We will update the final version of our paper with all the revisions discussed. We will also revise the typo issues you mentioned. We hope our responses address your concerns, and we are happy to answer any further questions you might have.
>
>
>
> $ $
>
>
> **Reference**
>
> [1] Li X, Dong G, Jin J, et al. Search-o1: Agentic search-enhanced large reasoning models[J]. arXiv preprint arXiv:2501.05366, 2025.
>
> [2] Jin B, Zeng H, Yue Z, et al. Search-r1: Training LLMs to reason and leverage search engines with reinforcement learning[J]. arXiv preprint arXiv:2503.09516, 2025.
>
> [3] Sheng G, Zhang C, Ye Z, et al. Hybridflow: A flexible and efficient RLHF framework[C]//Proceedings of the Twentieth European Conference on Computer Systems. 2025: 1279-1297.

---

> ### Author Response · Authors · 2025-08-04
> **Follow-up on Rebuttal**
>
> Dear reviewer zB4g,
>
>
> We truly appreciate your engagement with our work.
>
> To ensure we make the best use of the remaining discussion time, we would like to kindly check if our responses have clarified your main concerns. Please let us know if any part of our response is unclear or if there are any remaining concerns we can address. We would be more than happy to provide any further details or adjustments you might need.
>
> Thank you once again for your thoughtful engagement with our work.
>
> Best wishes,
>
>
>
> Authors

---

> > ### Comment · Reviewer_zB4g · 2025-08-08
> > **Reply to author's rebuttal**
> >
> > Thank you for your thorough rebuttal. It has addressed most of my concerns. With that in mind, I would like to increase my score from 4 to 5.

---

> > > ### Author Response · Authors · 2025-08-08
> > > **Thank you for your recognition**
> > >
> > > Dear Reviewer zB4g,
> > >
> > > Thank you for your recognition of our work. We sincerely appreciate your professional suggestions and kind response, which are valuable for improving our paper. We ensure that the points discussed in the rebuttal will be updated into the revised version.
> > >
> > > Once again, thank you for your thoughtful engagement. We are happy to provide any further details if needed.
> > >
> > >
> > > Best wishes,
> > >
> > > Authors

---

### Note · Authors · 2025-08-13

Dear Area Chair, Senior Area Chair, and Program Chairs,

We sincerely thank all reviewers for their thoughtful feedback and the Area Chair for their efforts in coordinating the review process, which has been invaluable in improving our work.

Most reviewers recognized the novelty of our framework, its solid theoretical analysis, and the comprehensive experimental validation. During the rebuttal stage, we provided detailed point-by-point responses to all reviewer suggestions, including clarifying key equations and addressing potential misunderstandings. We believe our responses have effectively addressed the concerns raised, and we will incorporate these clarifications and improvements into the final version of the paper.

Once again, we sincerely appreciate the reviewers and the Area Chair for their thoughtful engagement throughout the review process.

Best regards,

The Authors

---

### Decision · Program_Chairs · 2025-09-17

**Decision:**

Accept (poster)

**Comment:**

This paper proposes ExSearch, a framework that formulates iterative retrieval for LLMs as an EM-based optimization problem. The search process is cast as latent trajectories (think, search, record), optimized via iterative self-incentivization. The method combines trajectory sampling, importance weighting, and reweighted loss updates, offering both a practical agentic retrieval system and a theoretical convergence guarantee.
Extensive experiments across multiple datasets and model families show consistent improvements over strong RAG baselines. Additional results on GAIA and WebWalker further support generalization to more realistic, multi-step reasoning environments. Ablations, case studies, and efficiency comparisons provide further evidence of robustness.

Overall, the paper presents a principled, well-executed, and empirically validated framework for iterative retrieval in LLMs. The EM-based latent trajectory formulation offers both elegance and practical gains, with efficiency advantages over reinforcement learning–based approaches.

While there are valid critiques regarding novelty (XR4h) and overemphasis on mathematical formalism (zB4g), the majority of reviewers and AC agreed that the contribution is significant and relevant to the NeurIPS audience. The rebuttal effectively addressed concerns by clarifying baselines, adding GAIA and WebWalker results, and explaining theoretical vs. practical aspects more clearly. However, AC agrees the concern that the experiments all used on older datasets (e.g., HotpotQA), limiting claims of generalization. So I encourage authors to include a complete experimental table during camera-ready on GAIA and WebWalker results.